# Measuring Sparse Autoencoder Feature Space Similarities Across Large Language Models

## Abstract

The *Universality Hypothesis* in large language models (LLMs) claims that different models converge towards similar concept representations in their latent spaces. Providing evidence for this hypothesis would enable researchers to exploit universal properties, facilitating the generalization of mechanistic interpretability techniques across models. Previous works studied if LLMs learned the same *features*, which are internal representations that activate on specific concepts. Since comparing features across LLMs is challenging due to polysemanticity, in which LLM neurons often correspond to multiple unrelated features, rather to than distinct concepts, sparse autoencoders (SAEs) have been employed to disentangle LLM neurons into SAE features corresponding to distinct concepts. In this paper, we introduce a new variation of the universality hypothesis called *Analogous Feature Universality*: we hypothesize that even if SAEs across different models learn different feature representations, the spaces spanned by SAE features are similar, such that one SAE space is similar to another SAE space under rotation-invariant transformations. Evidence for this hypothesis would imply that interpretability techniques related to latent spaces, such as steering vectors, may be transferred across models via certain transformations. To investigate this hypothesis, we first pair SAE features across different models via activation correlation, and then measure spatial relation similarities between paired features via representational similarity measures, which transform spaces into representations that reveal hidden relational similarities. Our experiments demonstrate high similarities for SAE feature spaces across various LLMs, providing evidence for feature space universality.

## 1 Introduction

Large language models (LLMs) have demonstrated remarkable capabilities across a wide range of natural language tasks (Bubeck et al., 2023; Naveed et al., 2024). However, understanding how these models represent and process information internally remains a significant challenge (Bereska & Gavves, 2024). The *Universality Hypothesis* claims that different models converge towards similar concept representations in their latent spaces (Huh et al., 2024). Providing evidence for this hypothesis would enable researchers to exploit universal properties related to features, providing new ways for generalizing mechanistic interpretability techniques across models (Sharkey et al., 2024), and accelerating research towards safer and more controllable AI systems (Hendrycks et al., 2023).

If LLMs learn similar internal concept representations, called *features*, it is vital is to understand which measures they are similar under, and to quantify the extent of these similarities (Chughtai et al., 2023; Gurnee et al., 2024). Since comparing features across LLMs is challenging due to polysemanticity, in which individual neurons often correspond to multiple unrelated features, rather than to distinct concepts, sparse autoencoders (SAEs) have been employed to disentangle polysemantic LLM neurons into SAE features corresponding to distinct semantic concepts that are easier to analyze and compare across models (Cunningham et al., 2023). Previous works have found that while SAEs learn similar features, they also learn a notable percentage of different features that are idiosyncratic to specific SAEs (Leask et al., 2025; Paulo & Belrose, 2025). Even so, SAEs have been found to learn correlated features which activate on the same concepts across models (Bricken et al., 2023). Moreover, semantically similar SAE feature arrangements are found across SAEs trained on different models, such as "months" features arranged circularly in GPT-2, Mistral-7B, and Llama-3 8B (Engels et al., 2025; Leask et al., 2024). Research has also found other types of feature arrangements (Park et al.,

2024a; Li et al., 2025; Ye et al., 2024). These findings suggest investigating a new way to think of universality from the perspective of feature relations.

In this paper, we introduce a new variation of the universality hypothesis called **Analogous Feature Universality**: we hypothesize that even if SAEs across different models learn different feature representations, the spaces spanned by SAE features are similar, such that one SAE space is similar to another SAE space under certain rotation-invariant transformations. Evidence supporting this hypothesis would imply that latent spaces may have similar geometric arrangements of semantic subspaces (Templeton et al., 2024), suggesting that features, and techniques related to feature spaces (e.g. steering vectors), may be transferred across models via particular transformations. Previous works transfer features across models (Merullo et al., 2023), but there is a lack of research explaining why this can be done.

To investigate this hypothesis, we develop a novel approach that involves two main steps: (1) first, we pair SAE features across different models via activation correlation, and (2) afterwards, we measure the similarity of spatial relations between paired features using representational space similarity measures, which transform spaces into representations that reveal hidden relational similarities (Raghu et al., 2017; Kriegeskorte et al., 2008). This approach addresses a different research question compared to previous representational similarity papers (Kornblith et al., 2019; Klabunde et al., 2024; 2023; Sucholutsky et al., 2023), which **did not measure the relationships among features**, but among *input data samples* that were already known to correspond to the "same point" in input data space. In contrast, to investigate our hypothesis, our approach differs from previous approaches as it introduces step 1, which allows us to assess if we can use features, not input samples, to determine if representations are the same across models. Thus, we are applying representational similarity *based on features in weight space*, instead of based on input data samples in activation space. Appendix §I provides a simple example of an analogy to explain why step 1 is necessary in our approach: consistent high scores across models can only be found if these features are paired correctly.

Our experiments demonstrate high similarities in SAE feature spaces between middle layers across multiple LLMs, providing evidence for feature space universality. Furthermore, we show that semantically meaningful subspaces exhibit notable similarity across models.

**Our key contributions include:**

1. Empirical evidence for similar SAE feature space representations across diverse LLMs.

2. A novel approach to study similar feature spaces by comparing weight representations, instead of activation representations, via paired features.

3. An analysis of how feature similarity varies by semantic subspaces (e.g., Emotions).

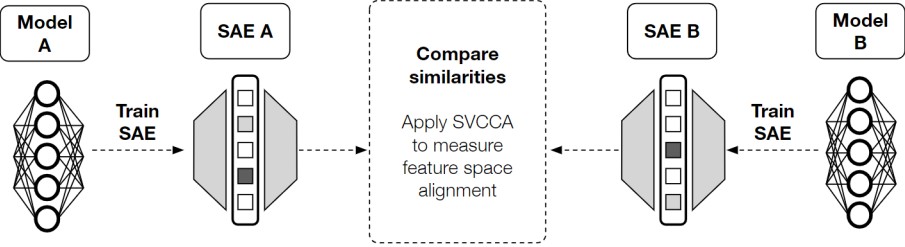

Figure 1: We train SAEs on LLMs, and then compare their SAE feature space similarity using measures such as Singular Value Canonical Correlation Analysis (SVCCA).

## 2 BACKGROUND AND DEFINITIONS

**Feature Universality.** We define a feature as an internal representation that activates on a concept (Park et al., 2024b). To quantify the extent of feature space universality, we utilize the following definitions that build on terms defined in previous works: We define **correlated features** as features which are correlated in terms of activation similarity, and thus activate highly on the same tokens

(Bricken et al., 2023). We measure the **representational universality** of feature spaces by applying representational similarity measures on a set of correlated feature pairs (Gurnee et al., 2024). Correlated features that reside in spaces with statistically significant representational similarity (as measured by the main approach discussed in Section §3) are labeled as **analogous features** (Olah et al., 2020). These similarity measures also measure how much *analogical similarity*, relative to a set of pairings and transformations, each feature has to its paired counterpart. Further discussion of these definitions is provided in Appendix §J.

We do not measure how well models capture **true features**, which are atomic ground truth features that models may converge towards capturing (Chughtai et al., 2023). Our hypothesis is that even if SAEs do not capture the same exact features, their features can be matched as *analogous features* that lie in spaces which can be transformed from one to another.

**Sparse Autoencoders.** Superposition is a phenomenon in which a model, in response to the issue of having to represent more features than it has parameters, learns feature representations distributed across many parameters (Elhage et al., 2022). This causes its parameters, or neurons, to be polysemantic, which means that each neuron is involved in representing more than one feature, making it difficult to compare features across different LLMs. To address this issue, Sparse Autoencoders (SAEs) have been applied to disentangle an LLM's polysemantic neuron activations into monosemantic "feature neurons", which are encouraged to represent an isolated concept.

SAEs are a type of autoencoder that learn sparse representations of input data by imposing a sparsity constraint on the hidden layer (Makhzani & Frey, 2013; Cunningham et al., 2023). An SAE takes in LLM layer activations $\mathbf{x} \in \mathbb{R}^n$ and reconstructs $\mathbf{x}$ as output $\hat{\mathbf{x}} = \mathbf{W}'\sigma(\mathbf{W}\mathbf{x} + \mathbf{b})$, where $\mathbf{W} \in \mathbb{R}^{h \times n}$ is the encoder weight matrix, $\mathbf{b}$ is a bias term, $\sigma$ is a nonlinear activation function, and $\mathbf{W}'$ is the decoder matrix, which often uses the transpose of the encoder weights. SAE training aims to both encourage sparsity in the activations $\mathbf{h} = \sigma(\mathbf{W}\mathbf{x} + \mathbf{b})$ and to minimize the reconstruction loss $\|\mathbf{x} - \hat{\mathbf{x}}\|_2^2$. The sparsity constraint encourages a large number of SAE neuron activations to remain inactive for any given input, while a few neurons, called *feature neurons*, are highly active. The active feature neurons tend to activate only for specific concepts in the data, promoting monosemanticity. Therefore, since there is a mapping from LLM neurons to SAE feature neurons that "translates" LLM activations to features, this method is a type of *dictionary learning* (Olshausen & Field, 1997). Several works make architectural improvements to SAEs (Gao et al., 2025; Rajamanoharan et al., 2024a;b), and analyze its ability to capture LLM features (Chanin et al., 2024).

*SAE Feature Weight Spaces.* By analyzing feature weights UMAPs of an SAE trained on a middle residual stream layer of Claude 3 Sonnet, researchers discovered feature spaces organized in neighborhoods of semantic concepts, and identified subspaces corresponding to concepts such as "biology" and "conflict" (Templeton et al., 2024). Additionally, these concepts appear to be organized in hierarchical clusters, such as "disease" clusters that contain sub-clusters of specific diseases like flus. In Section §4.3, we measure the extent of similarity of semantic feature spaces across LLMs.

**Representational Space Similarity.** Representational similarity measures typically compare neural network activations by assessing the similarity between activations from a consistent set of inputs (Klabunde et al., 2023). In this paper, we take a different approach by comparing the representations using the SAE decoder weight matrices $W'$, whose columns correspond to feature neurons. We calculate a similarity score $m(W'_X, W'_Y)$ for pairs of representations from SAEs $X$ We obtain two scores via the following representational similarity measures [1]:

*Singular Value Canonical Correlation Analysis (SVCCA)*: Given two sets of variables, $\mathbf{X} \in \mathbb{R}^{n \times d_1}$ and $\mathbf{Y} \in \mathbb{R}^{n \times d_2}$, Canonical Correlation Analysis (CCA) seeks to identify pairs of linear transformations $\mathbf{a}$ and $\mathbf{b}$ that maximize the correlation between $\mathbf{X}a$ and $\mathbf{Y}b$ (Hotelling, 1936). Singular Value Canonical Correlation Analysis (SVCCA) (Raghu et al., 2017) enhances CCA by first applying Singular Value Decomposition (SVD) to both $\mathbf{X}$ and $\mathbf{Y}$ to obtain the left singular matrices $\mathbf{U}_X$ and $\mathbf{U}_Y$ which form orthonormal bases for the column space of their matrices:

$$\mathbf{X} = \mathbf{U}_X \mathbf{S}_X \mathbf{V}_X^T, \quad \mathbf{Y} = \mathbf{U}_Y \mathbf{S}_Y \mathbf{V}_Y^T$$

---

[1]Appendix K discusses details on how these measures are applied on decoder weight matrices.

$\mathbf{S}_X$ and $\mathbf{S}_Y$ are diagonal matrices containing the singular values, and $\mathbf{V}_X^T$ and $\mathbf{V}_Y^T$ contain the right singular vectors. Dimensionality reduction is applied by truncating low variance directions, which reduces noise, to obtain $\mathbf{U}_{X,k}$ and $\mathbf{U}_{Y,k}$. These represent the principal directions of variance.

Then, CCA is applied on $\mathbf{U}_{X,k}$ and $\mathbf{U}_{Y,k}$ to find linear transformations $a$ and $b$ that maximize the Pearson correlation between $\mathbf{U}_{X,k}a$ and $\mathbf{U}_{Y,k}b$, such that $a$ and $b$ are the canonical directions in each space that are maximally correlated. The SVCCA score is the mean of the top canonical correlations. In other words, it measures how well linear relationships $\mathbf{U}_{X,k}a$ and $\mathbf{U}_{Y,k}b$ align.

*Representational Similarity Analysis (RSA)*: Representational Similarity Analysis (RSA) (Kriegeskorte et al., 2008) has been applied to measure relational similarity of stimuli across brain regions and of activations across neural network layers (Klabunde et al., 2023). First, for each space $\mathbf{X} \in \mathbb{R}^{n \times d_1}$ and $\mathbf{Y} \in \mathbb{R}^{n \times d_2}$, RSA constructs a Representational Dissimilarity Matrix (RDM) $\mathbf{D} \in \mathbb{R}^{n \times n}$, where each element of the matrix represents the dissimilarity (or similarity) between every pair of data points within a space. The RDM essentially summarizes the pairwise distances between all possible data pairs in the feature space. For space $\mathbf{X}$, the RDM has entries:

$$D_{ij}(X) = \|x_i - x_j\|_2,$$

where $x_i$ and $x_j$ are row vectors (e.g., features). A common distance metric used is the Euclidean distance. After an RDM is computed for each space, a correlation measurement like *Spearman's rank correlation coefficient* is applied to the pair of RDMs to obtain a similarity score.

## 3 METHODOLOGY

We compare SAEs trained on layer $A_i$ from LLM $A$ and layer $B_j$ from LLM $B$. This is done for every layer pair. To compare latent spaces using our selected measures, we have to solve both permutation and rotational alignment issues. For the permutation issue, we have to find a way to pair neuron weights. This was not an issue in previous representational similarity studies because activations were already paired by input data samples. (Klabunde et al., 2023). For our case, we do not pair activation vectors by input instances, but pair by feature neuron weights. This means we have to find neuron pairings in SAE $S_A$ to SAE $S_B$ based on individual *local similarity*. However, we do not know which features map to which features; the orderings of these features are permuted in the weight matrices, and some features may not have a corresponding partner. Therefore, we pair them via a correlation metric; a set of pairings is a *mapping*. For the rotational issue, models learn their own distinct basis for latent space representations, so features may be similar relation-wise across spaces, but not rotation-wise due to differing bases. To solve this, we employ rotation-invariant measures.

**Assessing Scores with Baseline.** We follow the approach of Kriegeskorte et al. (2008) to randomly pair features to obtain a baseline score. We compare the score of the features paired by correlation (which we refer to as the "paired features") with the average score of $N$ runs of randomly paired features to obtain a p-value score. If the p-value is less than 0.05, the measure suggests that the similarity is statistically meaningful. In summary, the steps to carry out our similarity comparison experiments are given below, and steps 1 to 3 are illustrated in Figure 19:

1. For the permutation issue: Get activation correlations for feature pairs from SAE decoder weights. Pair each SAE $A$ feature with its highest activation correlated feature from SAE $B$.

2. Rearrange the order of features in each matrix to pair them row by row.

3. For the rotational issue: Apply rotation-invariant measures to get a *Paired Score*.

4. Using the same measures, obtain the similarity scores of $N$ *random pairings* to estimate a null distribution. Obtain a p-value of where the paired score falls in the null distribution to determine statistical significance.

**Solving the Permutation Issue:** Due to learned parameter variations specific to each LLM and SAE, we do not expect every feature in a SAE to have a corresponding feature that is "close enough" in another SAE. Else, we would forcibly pair features without corresponding partners, yielding low scores due to "bad pairings". Thus, we compare SAE feature subsets.

*Highest Activation Correlation.* We take the activation correlations between every feature, following the approach of Bricken et al. (2023). We pass a dataset of samples through both models. Then, for each feature in SAE $A$, we find its activation correlation with every feature in SAE $B$, using tokens as the common instance pairing. Lastly, for each feature in SAE $A$, we find its highest correlated feature in SAE $B$ (as this allows for many-to-1 mappings, this is termed *1A-to-ManyB*; the other way around is *ManyA-to-1B*). We pair these features together when conducting our experiments. We refer to scores obtained by pairing via highest activation correlation as *Paired Scores*.

*Filter by Top Tokens.* We notice some pairings with "non-concept features" that have top 5 token activations on end-of-text / padding tokens, spaces, new lines, and punctuation; these non-concept pairings greatly reduce scores possibly due to being "bad pairings", as we describe further in Appendix M. By filtering out non-concept pairings, we significantly raise the similarity scores. We also only keep pairings that share at least one keyword in their top 5 token activations. The list of non-concept keywords is given in the Appendix M.

*Filter Many-to-1 Mappings.* We find that some mappings are many-to-1, meaning more than one feature maps to the same feature. We found that removing many-to-1 pairings slightly increased the scores, and we discuss possible reasons why this occurs in Appendix L. However, the scores still show high similarity even with the inclusion of many-to-1 pairings. We show scores of 1-to-1 pairings (filtering out many-to-1 pairings) in the main text, and show scores of many-to-1 pairings in Appendix L. We choose "many" features from the SAEs of the smaller LLM in a model pair, and find similar results if we choose from SAE of the larger LLM.

**Measures for the Rotational Issue**: Each rotation-invariant measure scores a type of *global, analogous similarity* under a feature mapping. We apply measures for: 1) How well subspaces align using **SVCCA**, and 2) How similar feature relations like king↔queen are using **RSA**. For RSA, we set the inner similarity function using Pearson correlation the outer similarity function using Spearman correlation, and compute the RDM using Euclidean Distance.

**Semantic Subspace Matching Experiments.** For these experiments, we first identify semantically similar subspaces, and then compare their similarity. We find a group of related features in each model by searching for features which highly activate on the top 5 samples' tokens that belong to a *concept category*. For example, the "emotions" concept contains keywords like "rage".

*Baseline Tests Types.* We use two types of tests which compare the paired score to a null distribution. Each test examines that the score is rare under a certain null distribution assumption.

1. Test 1: We compare the score of a semantic subspace mapping to the mean score of randomly shuffled pairings of the same subspaces. This test shows just comparing the subspace of features is not enough; the features must be paired.

2. Test 2: We compare the score of a semantic subspace mapping to a mean score of mappings between randomly selected feature subsets of the same sizes as the semantic subspaces. This shows that the high similarity score does not hold for any two subspaces of the same size.

## 4 EXPERIMENTS

### 4.1 EXPERIMENTAL SETUP

**LLM Models.** We compare LLMs that use the Transformer model architecture (Vaswani et al., 2017). We compare models that use the same tokenizer because the highest activation correlation pairing relies on comparing two activations using the same tokens. In the main text, we compare the following residual stream layers pairs: (1) Pythia-70m, which has 6 layers and 512 dimensions, to Pythia-160m, which has 12 layers and 768 hidden dimensions (Biderman et al., 2023), (2) Gemma-1-2B, which has 18 layers and 2048 dimensions, to Gemma-2-2B, which has 26 layers and 2304 dimensions (Team et al., 2024a;b), and (3) Gemma-2-2B to Gemma-2-9B, which has 42 layers and 3584 dimensions. Appendix §D compares Llama 3-8B-Instruct and Llama 3.1-8B. In total, we compare 8 model pairs, as listed in Appendix §A.

**SAE Models.** For Pythia models, we use SAEs with 32768 feature neurons trained on residual stream layers from Eleuther (EleutherAI, 2023). For the Gemma models, we use SAEs with 16384 feature neurons trained on residual stream layers (Lieberum et al., 2024). We access pretrained

Gemma and Llama SAEs through the SAELens library (Bloom & Chanin, 2024). In Appendix §C, we compare SAEs for base vs fined tuned versions for Gemma-2-9B vs Gemma-2-9B-Instruct and Gemma-1-2B vs Gemma-1-2B-Instruct (at Layer 12). In Appendix §E, we assess a baseline comparison that compares Pythia-70m SAEs with SAEs trained on a randomized Pythia-70m. In Appendix §F, we compare SAEs trained within the same model. In Appendix §G, we compare SAEs across Pythia-70m and Pythia-160m as we vary SAE widths.

**Datasets.** We obtain SAE activations using OpenWebText, a dataset that scraps internet content (Gokaslan & Cohen, 2019). We use 100 samples with a max sequence length of 300 for Pythia for a total of 30k tokens, and we use 150 samples with a max sequence length of 150 for Gemma for a total of 22.5k tokens. Top activating dataset tokens were also obtained using OpenWebText samples. In Appendix §H, we vary the dataset and number of tokens used to obtain activations to show that our results do not drastically change when these factors are varied.

**Computing Resources.** We run experiments on an A100 GPU.

### 4.2 SAE Space Similarity

In summary, for almost all layer pairs after layer 0, we find the p-value of SAE experiments for SVCCA, which measures rotationally-invariant alignment, to be low; most are between 0.00 to 0.01 using 100 samples.[2] While RSA also passes many p-value tests, their values are much lower, suggesting a relatively weaker similarity in terms of pairwise differences. It is possible that many features are not represented in just one layer; hence, a layer from one model can have similarities to multiple layers in another model. We also note that the first layer of every model, layer 0, has a very high p-value when compared to every other layer, including when comparing layer 0 from another model. This may be because layer 0 contains few discernible, meaningful, and comparable features. We find that middle layer pairs have the highest scores. On the contrary, early layers have much lower scores, perhaps due to lacking content, while later layers also have scores lower than middle layers, perhaps due to being too specific/complex.

Because LLMs, even with the same architecture, can learn many different features, we do not expect entire feature spaces to be highly similar; instead, we hypothesize there may be highly similar *subspaces*. In other words, there are many similar features universally found across SAEs, but not every feature learned by different SAEs is the same. Our results support this hypothesis. For all experiments in this section, approximately 10-30% (and typically 20%) of feature pairs are kept after filtering both non-concept and Many-1 pairings.

*Activation Correlation.* As shown in Figure 24 for Pythia and Figure 25 for Gemma-1 vs Gemma-2 in Appendix O, we find that mean of the highest (local) activation correlations does not always correlate with the global similarity measure scores. For instance, a subset of feature pairings with a moderately high mean activation correlation (eg. 0.6) may yield a low SVCCA score (eg. 0.03).

**Pythia-70m vs Pythia-160m.** In Figure 2, we compare Pythia-70m, which has 6 layers, to Pythia-160m, which has 12 layers. When compared to Figures 26 and 27 in Appendix O, for all layers pairs (excluding Layers 0), and for both SVCCA and RSA scores, almost all the p-values of the residual stream layer pairs are between 0% to 1%, indicating that the feature space similarities are statistically meaningful. In particular, Figure 2 shows that Layers 2 and 3, which are middle layers of Pythia-70m, are more similar by SVCCA and RSA scores to middle layers 4 to 7 of Pythia-160m compared to other layers. Figure 20 also shows that Layer 5, the last layer of Pythia-70m, is more similar to later layers of Pythia-160m than to other layers. The number of features after filtering non-concept and many-to-1 features is given in Tables 5 and 6 in Appendix O.

**Gemma-1-2B vs Gemma-2-2B.** As shown by the paired SVCCA and RSA results for these models in Figure 3, we find that using SAEs, we can detect highly similar features in layers across Gemma-1-2B and Gemma-2-2B. Compared to mean random pairing scores and p-values in Figures 28 and 29 in Appendix O, for all layers pairs (excluding Layers 0), and for both SVCCA and RSA scores, almost all the p-values of the residual stream layer pairs are between 0% to 1%, indicating that the feature space similarities are statistically meaningful. The number of features after filtering non-concept features and many-to-1 features are given in Tables 7 and 8 in Appendix O.

---

[2]Only 100 samples were used as we found the variance to be low for the null distribution, and that the mean was similar for using 100 vs 1k samples.

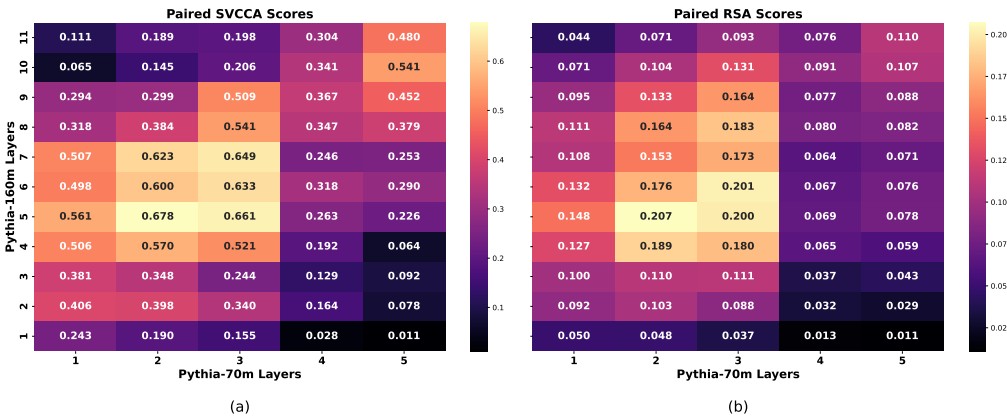

(a)                                    (b)

Figure 2: (a) SVCCA and (b) RSA 1-1 paired scores of SAEs for layers in Pythia-70m vs layers in Pythia-160m. We find that middle layers have the most similarity with one another (as shown by the high-similarity block spanned by layers 1 to 3 in Pythia-70m and Layers 4 to 7 in Pythia-160m). We exclude layers 0, as we observe they always have non-statistically significant similarity. The 1-1 scores are slightly higher for most of the Many-to-1 scores shown in Figure 20, and the SVCCA score at L2 vs L3 for 70m vs 160m is much higher.

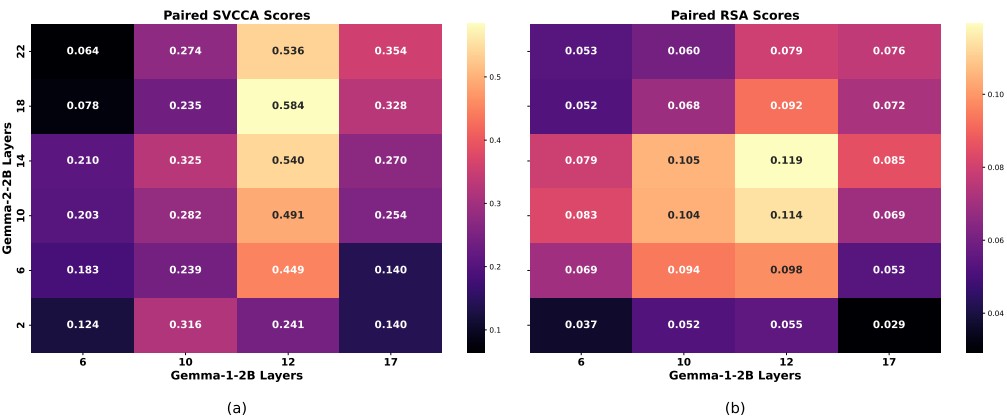

(a)                                    (b)

Figure 3: (a) SVCCA and (b) RSA 1-1 paired scores of SAEs for layers in Gemma-1-2B vs layers in Gemma-2-2B. Middle layers have the best performance. The later layer 17 in Gemma-1 is more similar to later layers in Gemma-2. Early layers like Layer 2 in Gemma-2 have very low similarity. We exclude layers 0, as we observe they always have non-statistically significant similarity. The 1-1 scores are slightly higher for most of the Many-to-1 scores shown in Figure 22.

**Gemma-2-2B vs Gemma-2-9B.** We obtain activations from OpenWebText using 150 samples with a max sequence length of 150, for a total of 22.5k tokens. Figure 4 shows SVCCA and RSA scores after filtering keywords and for 1-1. In particular, the L11 vs L21 comparison achieves an SVCCA score of 0.70 (while mean random pairing has a score of 0.009) and an RSA score of 0.195 (while mean random pairing for has a score of $4.38 \times 10^{-4}$). For these experiments, we use 10 runs for mean random pairings at only one layer pair due to the longer run times for larger model pairs.

### 4.3 SIMILARITY OF SEMANTICALLY MATCHED SAE FEATURE SPACES

We find that for all layers and for all concept categories, Test 2 described in §3 is passed. Thus, we only report specific results for Test 1 in Tables 1 and 2. Overall, in both Pythia and Gemma models and for many concept categories, we find that semantic subspaces are more similar to one another than non-semantic subspaces. More discussion on these results is given in Appendix N.

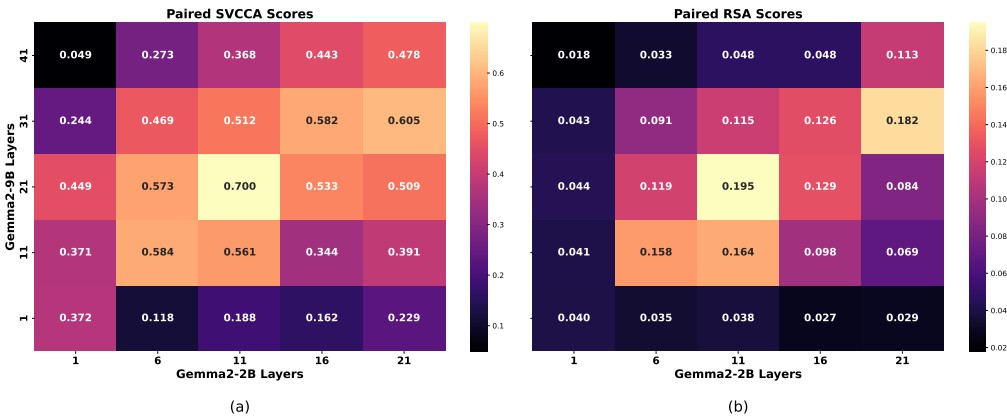

Figure 4: (a) SVCCA and (b) RSA 1-1 paired scores of SAEs for layers in Gemma-2-2B vs layers in Gemma-2-9B. Middle layers have the best performance. We exclude layers 0, as we observe they always have non-statistically significant similarity.

Table 1: SVCCA scores and random mean results for 1-1 semantic subspaces of L3 of Pythia-70m vs L5 of Pythia-160m. P-values are taken for 1000 samples in the null distribution.

| Concept Subspace | Number of Features | Paired Mean | Random Shuffling Mean | p-value |
|:---:|:---:|:---:|:---:|:---:|
| Time | 228 | 0.59 | 0.05 | 0.00 |
| Calendar | 126 | 0.65 | 0.07 | 0.00 |
| Nature | 46 | 0.50 | 0.12 | 0.00 |
| Countries | 32 | 0.72 | 0.14 | 0.00 |
| People/Roles | 31 | 0.50 | 0.15 | 0.00 |
| Emotions | 24 | 0.83 | 0.15 | 0.00 |

Rather than just finding that middle layers are more similar to middle layers, we find that they are the best at representing concepts. This is consistent with work by Rimsky et al. (2024), which find that middle layers work best for steering behaviors such as sycophancy. Other papers find that middle layers tend to provide better directions for eliminating refusal behavior (Arditi et al., 2024) and work well for representing goals and finding steering vectors in policy networks (Mini et al., 2023).

**Pythia-70m vs Pythia-160m.** We compare every layer of Pythia-70m to every layer of Pythia-160m for several concept categories. While many layer pairs have similar semantic subspaces, middle layers appear to have the semantic subspaces with the highest similarity. Table 1 demonstrates one example of this by comparing the SVCCA score for layer 3 to layer 5 of Pythia-160m, which shows high similarity for semantic subspaces across models, as the concept categories all pass Test 1, having p-values below 0.05. We show the scores for other layer pairs and categories in Figures 30 and 32 in Appendix O, which include RSA scores in Table 9.

**Gemma-1-2B vs Gemma-2-2B.** In Table 2, we compare L12 of Gemma-1-2B vs L14 of Gemma-2-2B. As shown in Figure 22, this layer pair has a very high similarity for most concept spaces; as such, they likely have high similar semantically-meaningful feature subspaces. Notably, not all concept group are not highly similar; for instance, unlike in Pythia, the Country concept group does not pass the Test 1 as it has a p-value above 0.05. We show the scores for other layer pairs and categories in Figures 34 and 36 in Appendix O.

## 5 RELATED WORK

**Feature Universality.** Previous works studied universality in terms of components such as features and circuits (Olah & Batson, 2023; Chughtai et al., 2023). Bricken et al. (2023) measure individual SAE feature similarity for two 1-layer toy models; however, this study did not analyze the global

Table 2: SVCCA scores and random mean results for 1-1 semantic subspaces of L12 of Gemma-1-2B vs L14 of Gemma-2-2B. P-values are taken for 1000 samples in the null distribution.

| Concept Subspace | Number of Features | Paired SVCCA | Random Shuffling Mean | p-value |
|---|---|---|---|---|
| Time | 228 | 0.46 | 0.06 | 0.00 |
| Calendar | 105 | 0.54 | 0.07 | 0.00 |
| Nature | 51 | 0.30 | 0.11 | 0.01 |
| Countries | 21 | 0.39 | 0.17 | 0.07 |
| People/Roles | 36 | 0.66 | 0.15 | 0.00 |
| Emotions | 35 | 0.60 | 0.12 | 0.00 |

properties of feature spaces. Gurnee et al. (2024) find evidence of universal neurons across language models by applying pairwise correlation metrics. Ye et al. (2024) discover "feature families" representing related hierarchical concepts in SAEs across semantically different datasets. Huh et al. (2024) showed that as vision and language models are trained with more parameters and with better methods, their representational spaces converge towards more similar representations. Crosscoders (Lindsey et al., 2024; Mishra-Sharma et al., 2025) and Universal SAEs (Thasarathan et al., 2025) are SAE variations that capture individual features shared across models.

**Representational Space Similarity.**    Previous work has studied neuron activation spaces by utilizing metrics to compare the geometric similarities of representational spaces (Raghu et al., 2017; Wang et al., 2018b; Kornblith et al., 2019; Klabunde et al., 2024; 2023; Kriegeskorte et al., 2008; Sucholutsky et al., 2023), finding that even models with different architectures may share similar representation spaces, hinting at feature universality. However, these techniques have yet to be applied to the feature spaces of SAEs trained on LLMs. Moreover, these techniques compare the similarity of paired input activations. Our work differs as it compares the similarity of paired *feature weights*.

**Mechanistic Interpretability.**    Previous work has made notable progress in neuron interpretation (Foote et al., 2023; Garde et al., 2023) and interpreting the interactions between components (Neo et al., 2024). Other work in mechanistic interpretability focus on circuits analysis (Elhage et al., 2021), as well using steering vectors to control model behaviors (Zou et al., 2023; Turner et al., 2024). These approaches have been combined with SAEs to steer models in the more interpretable SAE feature space (Chalnev et al., 2024). Steering vectors have been found across models, such as vectors that steered a model to refuse harmful instructions across 13 different models (Arditi et al., 2024). SAEs can also find steerable safety-related features across models, such as refusal features (O'Brien et al., 2024). Finding universal control methods can assist with AI safety issues (Barez et al., 2023).

# 6    CONCLUSION

We take the first steps in investigating the unexplored topic of feature space universality in LLMs via SAEs trained on different models. To achieve these insights, we develop a novel methodology to pair features with high activation correlations, and assess the global similarity of the resulting subspaces using SVCCA and RSA. Our findings reveal a high degree of SVCCA similarity in SAE feature subspaces across various models, particularly between their middle layers. Furthermore, our research reveals that certain subspaces of features associated with semantic categories, such as calendar or people tokens, demonstrate high similarity across different models. This suggests that certain semantic feature subspaces may be universally encoded across varied LLM architectures. Lastly, while we discover evidence that LLMs learn "weakly universal" features, and that SAEs can capture both these features and these relations, we also find that there is a notable fraction of features idiosyncratic to different SAEs, a finding that is supported by previous work (Leask et al., 2025; Paulo & Belrose, 2025). Thus, our research guides future work to improve upon methods that can better capture universal feature representations, and contributes to ongoing research that aims to improve SAEs (Engels et al., 2024).

## REPRODUCIBILITY

We include simple notebooks for reproducing the main results of Figures 2 and 3, and Tables 1 and 2, in the Supplementary Materials. Further code for reproducibility can be made available upon request.

## USE OF LLMS IN WRITING

We use LLMs for minor writing assistance and to aid in finding related work.

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

## A  LIST OF MODELS ANALYZED IN EXPERIMENTS

1. Pythia-70m VS Pythia-160m
2. Gemma-1-2B VS Gemma-2-2B
3. Gemma-2-2B vs Gemma-2-9B
4. Gemma-2-9B vs Gemma-2-9B-Instruct
5. Gemma-1-2B vs Gemma-1-2B-Instruct (at Layer 12)
6. Llama 3-8B and Llama 3.1-8B (at Layer 25)
7. Pythia-70m vs Randomized Pythia-70m (*Baseline Comparisons*)
8. TinyStories-1L-21M (*Within Model Comparisons*)

Unless otherwise specified, all experiments filter feature pairs by non-concept keywords, 1-1, and low correlation (less than 0.1).

## B  TRAINED SAE METRIC EVALUATIONS

For SAEs with an expansion factor of 64 (and thus 32768 total latents) and a top-k of 32 trained on residual stream layers of Pythia-70m using 100 million tokens from the RedPajama dataset, we obtain explained variances of between 0.70 to 0.82 ($\mu = 0.75, \sigma = 0.04$), cosine similarities between 0.92 to 0.99, cross entropy losses between 0.78 to 0.94, and L1 norm between 24.0 to 50.75 ($\mu = 35.55, \sigma = 8.78$). Likewise, these are also SAEs with good reconstruction and sparsity metrics.

In general, all of our custom trained SAEs on Pythia-70m and on Pythia-160m have similar metrics as the ones reported for the SAEs in this section. We evaluate the SAEs on SAEBench (Karvonen et al., 2024).

For the random SAEs trained on identical conditions as the non-random SAEs mentioned above, but on a randomized Pythia-70m with the embedding layer not being randomized, we obtain explained variances of between 0.63 to 0.80 ($\mu = 0.71, \sigma = 0.006$), cosine similarities 1, cross entropy losses of 1, and L1 norm between 27.38 to 38.25. These results are similar to the results for Pythia-70m-deduped SAEs found by Heap et al. (2025), and indicates SAEs with good reconstruction and sparsity metrics.

However, for our experiments that use these custom trained SAEs, we find that only around 2 to 4% of feature pairs are kept on average, in contrast to the 20 to 30% of features kept when running the pretrained models hosted by Eleuther and SAELens. We attribute this low amount of feature spaces to the quality of our trained SAEs in capturing features; while SAEs like those trained by Eleuther used around 8 billion tokens, our SAEs were trained using 100 million tokens, and as such may not be capturing features as well.

## C    BASE VS FINE TUNED RESULTS

### C.1    GEMMA-2-2B VS GEMMA-2-9B-INSTRUCT RESULTS

We compare Gemma-2-2B vs Gemma-2-9B-Instruct, a fine tuned version, at Layers 9, 20, and 31 of each model. Each SAE has a width of 16384 features. Only the pretrained SAEs for layers 9, 20 and 31 were publically available online and hosted on SAELens. We obtain activations from the RedPajamas dataset (Weber et al., 2024) using 150 samples with a max sequence length of 150, for a total of 22.5k tokens. On average, 22.2% of feature pairs are kept after filtering, with the range between 16.4% and 33.5%. Figure 5 shows that closer layers had higher SVCCA scores, while RSA scores were low except for matching layers. This is consistent with previous work which found that SAE features can transfer across base to fine-tuned language models (Kissane et al., 2024a; Kutsyk et al., 2024).

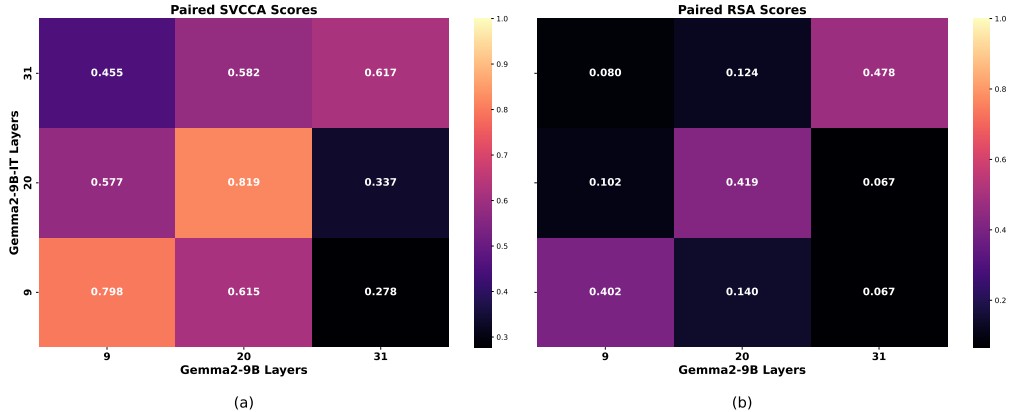

Figure 5:   Gemma-2-9B vs Gemma-2-9B-Instruct 1-1 paired SAE scores for (a) SVCCA and (b) RSA.

We also compare Gemma-1-2B vs Gemma-1-2B-Instruct, a fine tuned version, at Layer 12 of each model. Each SAE has a width of 16384 features. We obtain activations using 150 samples with a max sequence length of 150, for a total of 22.5k tokens. After filtering, the comparison achieves an SVCCA score of 0.84 (while mean random pairing has a score of 0.008) and an RSA score of 0.25 (while mean random pairing has a score of $4.29 \times 10^{-4}$). Around 50% of feature pairs are kept after filtering.

## D    LLAMA 3-8B-INSTRUCT VS LLAMA 3.1-8B RESULTS

We compare Llama 3-8B-Instruct and Llama 3.1-8B at each model's Layer 25 residual stream, a later layer in both models. The Llama 3 SAE has a width of 65536 features and uses Gated ReLU (Rajamanoharan et al., 2024a), while the Llama 3.1 SAE has a width of 32768 features and uses JumpReLU (Rajamanoharan et al., 2024b). The SAELens library (Bloom & Chanin, 2024) only hosted one SAE for Llama 3 at the time of our work.

We obtain activations using 100 samples with a max sequence length of 100, for a total of 10000 tokens. After filtering to keep 7% of Llama 3.1-8B feature pairs, this comparison achieves an SVCCA score of 0.3, comparable to the score of later layers in both Gemma and Pythia, and a mean activation correlation score of 66%.

## E    BASELINE COMPARISONS TO SAEs TRAINED ON LLMs WITH RANDOMIZED WEIGHTS

Recent research has demonstrated that training SAEs on randomly initialized transformers—where parameters are independently sampled from a Gaussian distribution rather than learned from text

data—yields latent representations that are similarly interpretable to those from trained transformers (Heap et al., 2025). One caveat is that the interpretable features of SAEs trained on "randomized" LLMs tend to activate on simple features, such as ones that activate on specific single tokens instead of more general concepts, rather than complex abstract features learned by middle to later layer SAEs. The authors propose that there may be cases where "simple" interpretable latents may arise from the inherent sparsity of the text data used to train language models, rather than from uncovering the computational structure embedded within the underlying model. Thus, we test whether the feature similarities found across LLMs are due to the sparsity of the training data or due to the computational structure. We run experiments comparing the representational similarity of SAE features trained on randomized LLMs to SAE trained on the original LLMs. We call the former "random SAEs", and the latter "non-random SAEs".

We find that, relative to other across model experiments, there is little similarity between the non-random SAEs and random SAEs, which suggests that our method is demonstrating SAE feature space similarity that is capturing the model's computational structure, and is not just capturing similarity due to interpretability from the inherent sparity of the text data. In particular, not even the same layers have medium similarity. However, the small amounts of similarity do suggest that SAEs trained on even randomized LLMs can capture features in the data, which is consistent with the interpretable "simple, low information" features, such as those that just activate on single tokens, that were also present in random SAEs found by Heap et al. (2025).

**SAE Models.** For Pythia-70m, we compare residual stream layer SAEs trained on a randomized Pythia-70m, to SAEs trained on the original Pythia-70m. Both SAEs were trained using 100 million tokens from the RedPajama dataset (Weber et al., 2024), with an expansion factor of 64 and a Top-K of 32. The embedding layer was not randomized. We evaluate the SAEs on SAEBench (Karvonen et al., 2024) and report their results in Section §B. Overall, all SAEs obtained good reconstruction and sparsity metric scores.

**Results.** As shown in Figure 6, using 200 samples with a max sequence length of 200 for a total of 40k tokens, we find that after filtering, we obtain very low SVCCA and RSA scores, indicating that random SAEs do not find features with notably similar relations compared to non-random SAEs. In comparison, the non-random SAEs obtained higher similarity scores when compared to other SAEs also trained on Pythia-70m, such as when comparing to the Eleuther SAEs as shown in Figure 12. Additionally, we find that only 1% of feature pairs are kept on average. The reason why this may occur is discussed in Section §B.

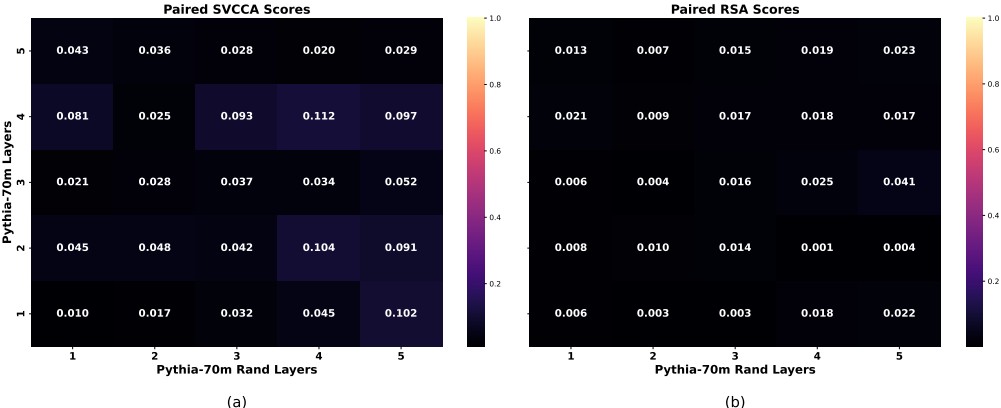

Figure 6: Baseline Comparisons with SAEs trained on Randomized Pythia-70m: (a) SVCCA 1-1 paired scores of SAEs, (b) RSA 1-1 paired scores of SAEs.

# F WITHIN MODEL FEATURE SPACE SIMILARITY RESULTS

## F.1 COMPARING THE SAME SAE TO THEMSELVES

As a sanity check that our method works, we compare running the same Pythia SAEs against themselves using our method. As expected, the same layers achieve 99 to 100% similar activation correlation, SVCCA, and RSA scores. However, for other layer pairs, our results obtained for the same model demonstrate that similarity scores greatly depend on the quality of the SAE that is trained. As shown in Figure 7, the Pythia-70m 32k latent SAEs trained on 8.2 billion tokens from The Pile show notable similarity between middle and later layers. However, as shown in Figure 8, the Pythia-70m 32k latent SAEs trained on only 100 million tokens from RedPajamas, which were trained on 1% of the training data as Eleuther's SAEs, show almost no similarity for some middle / later layer pairs. This demonstrates that while models may have notable similarity, these similarities may not always be revealed by just any SAE; the quality of the SAE in capturing these features is important. Additionally, different SAEs may reveal different features. For instance, the Layer 5 to Layer 1 pair in Figure 7 shows low similarity, while the same pair in Figure 8 shows medium-high similarity.

This is consistent with recent results which found that SAEs do not learn consistent features (Leask et al., 2025; Hindupur et al., 2025). Thus, while we show that certain subspaces learned by LLMs have high similarity, as revealed by SAEs, these subspaces may not always be consistently captured by arbitrary SAEs. This demonstrates a limitation of SAEs- they can capture different features that are shared across models, but cannot do this consistently. Thus, future work can attempt to address this limitation to develop SAEs that capture features more consistently. Indeed, recent work has already developed "universal SAEs" that can better capture features found across models (Thasarathan et al., 2025).

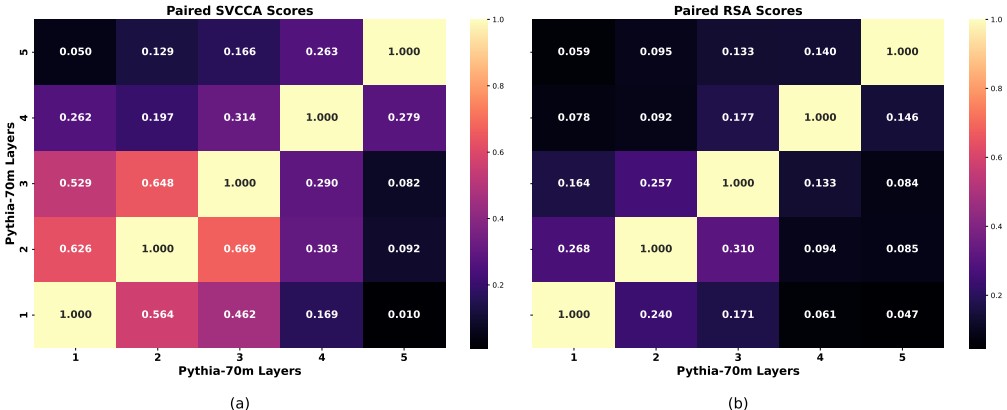

(a)                                                                 (b)

Figure 7: (a) SVCCA 1-1 paired scores of the same SAE trained on 8.2 billion tokens from The Pile by Eleuther, (b) RSA 1-1 paired scores of the same SAE.

## F.2 COMPARING SAES TRAINED WITH DIFFERENT SEEDS

**Pythia-70m Results .** We trained two SAEs with different seeds on Pythia-70m. As discussed in Section §F.1, due to training limitations such as only using 100 million tokens during training, these SAEs do not capture features as well as higher quality SAEs such as Eleuther's SAEs trained on 8.2 billion tokens from the Pile. Thus, we compare the results of SAEs trained on different seeds with the results of Figure 8, which compare the SAE with itself as a baseline, rather than assuming that the adjacent layers must be highly similar (or in other words, Figure 7 would not be suitable as a baseline).

The SAE results on different seeds is shown in Figure 9, and show some resemblance to the results of Figure 8. Notably, this occurs in how the first layer (bottom row) have high similarities, while the later layer comparisons show very low or no similarity. However, there are still notable dissimilarities; for instance, the same-layer comparisons along the diagonal for SAEs trained on different seeds only

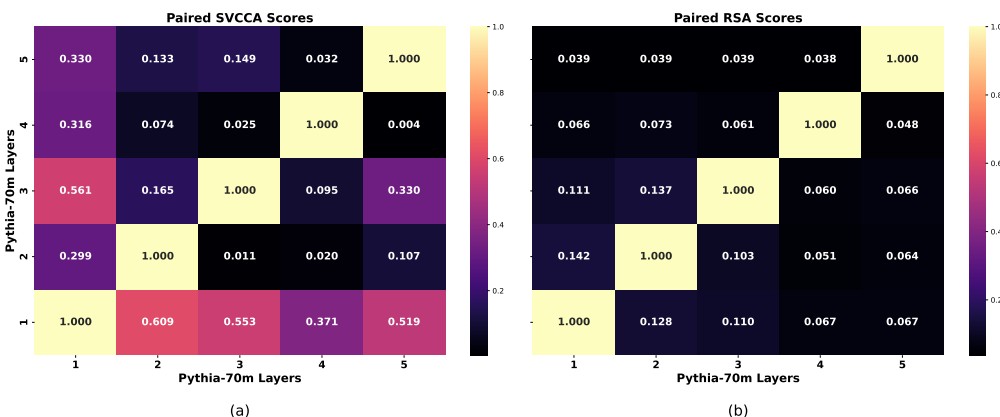

Figure 8: (a) SVCCA 1-1 paired scores of the same SAE trained on 100 million tokens from RedPajamas, (b) RSA 1-1 paired scores of the same SAE.

have a similarity of 0.376 on average, which is much less than the similarity of 1 for the same SAEs. Thus, the results of comparing SAEs trained on different seeds are similar to the results of comparing the same SAE to itself, but with notable differences.

As mentioned before in Section §F.1, this is consistent with recent results which found that SAEs do not learn consistent features (Leask et al., 2025; Hindupur et al., 2025). Despite not finding the same features, which can be viewed as a form of strong universality, these SAEs still show notable similarity. This can be viewed as a form of weak universality; see Appendix J for further discussion on this topic.

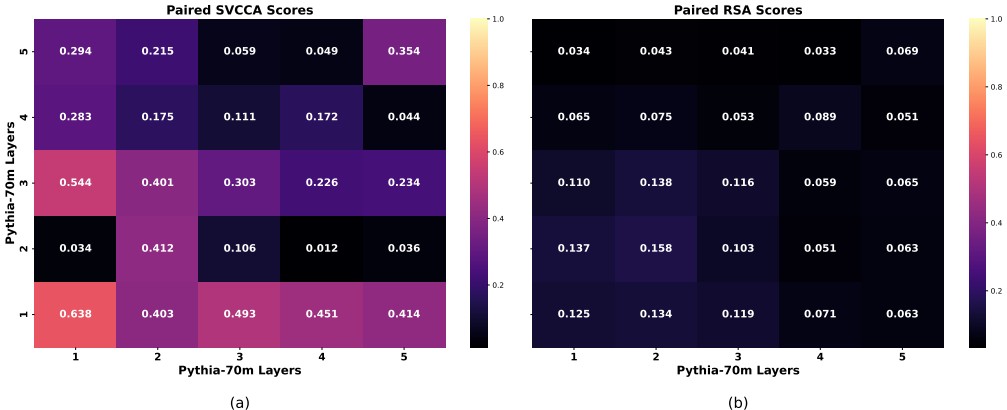

Figure 9: (a) SVCCA 1-1 paired scores of two SAEs trained with different seeds on 100 million tokens from RedPajamas, (b) RSA 1-1 paired scores of the SAEs. The SAE on the rows (y-axis) is the same SAE as the one in Figure 8.

**Tinystories Results .** To examine SAE feature space similarities when trained on the same model, we train multiple SAEs on a one-layer Tinystories model.

*SAE Training Parameters.* We train two SAEs at different seeds for a TinyStories-1L-21M model (Eldan, 2023). Both SAEs are trained at 30k steps with expansion factor of 16, for a 16384 total features, on a TinyStories dataset, and using ReLU as the activation function. We use a batch size of 4096 tokens and train with an Adam optimizer, applying a constant learning rate of $1 \times 10^{-5}$. The learning rate follows a warm-up-free schedule with a decay period spanning 20% of the total training steps. L1 sparsity regularization is applied with a coefficient of 5, and its effect gradually increases over the first 5% of training steps. We obtain SAE activations using 500 samples from the TinyStories dataset with a max sequence length of 128 (for a total of 64000 tokens).

*ManyA-to-1B.* This experiment allows multiple features from SAE $A$ to map onto a single feature from SAE $B$. Before filtering non-concept features, this comparison achieves an SVCCA score of 0.73, retaining 9930 feature pairs. After filtering for non-concept features (for this study, we only filter for periods and padding tokens), this pair achieves an SVCCA score of 0.94. Randomly paired achieves a score of 0.004 for 10 runs. We found that filtering for 1-1 did not change the score much. In contrast, for 10 random runs where we randomly choose 7 tokens that are not part of the non-concept tokens, the filtering of these randomly chosen tokens does not change the unfiltered SVCCA score of 0.73.

*1A-to-ManyB.* This experiment allows multiple features from SAE $B$ to map onto a single feature from SAE $A$. Before filtering nonconcept features, this comparison achieves an SVCCA score of 0.8. After filtering for nonconcept features, this pair achieves an SVCCA score of 0.94, retaining 13073 feature pairs. We found that filtering for 1-1 did not change the score much. The results for "randomly chosen tokens filtering" and "randomly shuffled pairing" match those for the ManyA-to-1B experiment.

*Cosine Similarity.* Given that these SAEs were trained on the same model, we also compare use cosine similarity instead of activation correlation to pair the features. We obtain an average cosine similarity of 0.9. When using cosine similarity to pair the SAE features, we obtain an SVCCA score of 0.92. This occurs whether we use 1A-to-ManyB or ManyA-to-1B. Every feature finds a unique pairing, despite neither matrix containing exact matches for their features with one another.

*Different SAE Widths.* We find similar results for SAEs trained using an expansion factor of 8, for 8192 total features. For ManyA-to-1B, before filtering, an SVCCA score of 0.84 is achieved, and after filtering, an SVCCA score of 0.94 is achieved. For 1A-to-ManyB, before filtering, an SVCCA score of 0.87 is achieved, and after filtering, an SVCCA score of 0.94 is achieved.

## F.3 COMPARING SAEs WITH DIFFERENT WIDTHS

Figure 10 shows SAEs with 32k latents vs SAEs with 16k latents, both trained on Pythia-70m. This 32k SAE is the SAE shown on the columns (x-axis) of Figure 9. After filtering, around 3.47% of feature pairs are kept on average. Figure 11 shows an SAE with 64k latents vs an SAE with 32k latents, both trained on Pythia-70m. After filtering, around 3.88% of feature pairs are kept on average. The reason why this may occur is discussed in Section §B.

In general, we do not find noticeable trends in scores as we vary SAE widths. This may be because each SAE varies in the features they capture Leask et al. (2025), so while they capture feature similarities across models, the similar features that they do capture are not the same for each SAE. We did not evaluate for feature splitting or absorption (Chanin et al., 2024).

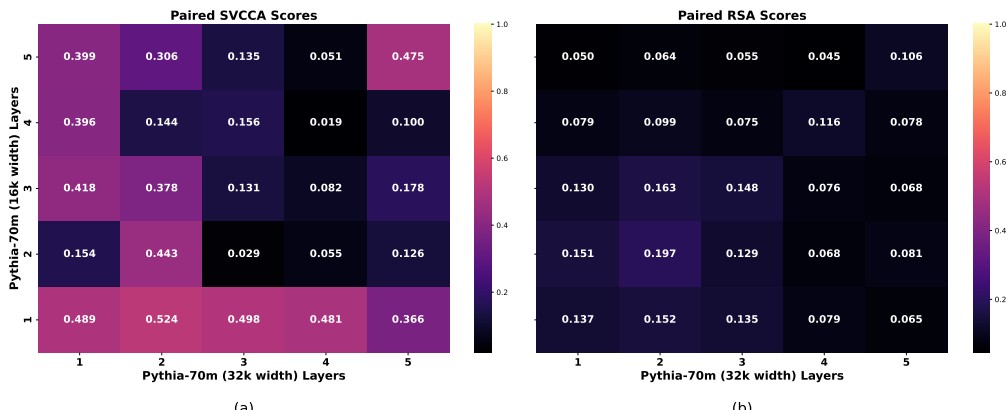

Figure 10: (a) SVCCA 1-1 paired scores of an SAE with 32k latents vs an SAE with 16k latents, (b) RSA 1-1 paired scores of the SAEs.

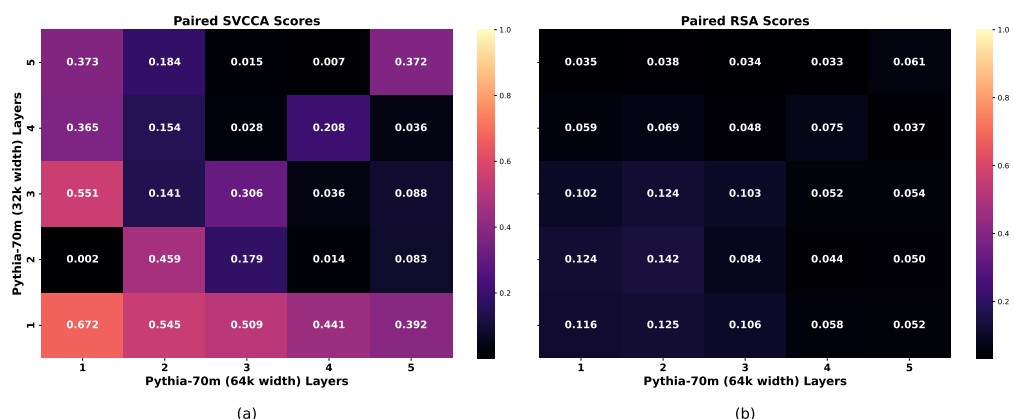

(a)                                              (b)

Figure 11: (a) SVCCA 1-1 paired scores of an SAE with 64k latents vs an SAE with 32k latents, (b) RSA 1-1 paired scores of the SAEs.

## F.4 COMPARING SAEs TRAINED ON DIFFERENT DATASETS

We compare SAEs on Pythia-70m trained on different training data. We trained an SAE trained on the residual stream layers of Pythia-70m using 100 million tokens using the RedPajama dataset at a context length of 256, with 32768 feature neurons (an expansion factor of 64 and Top-K) of 32. We compare our SAE with Eleuther's SAE trained at residual streams using 8.2 billion tokens from the Pile training set at a context length of 2049 with 32768 feature neurons and Top-K of 16.

To obtain activations, we use 200 samples with a max sequence length of 200, and after filtering around 4.58% of feature pairs are kept on average, in contrast to the 20 to 30% of feature pairs kept by comparing Eleuther's pretrained SAEs with each other, perhaps due to the lower quality of our SAEs as discussed in Section §B.

Figure 12 shows that the two sets of SAEs show some similarities, such as at Layers 1, 3, and 5, but also notable dissimilarities, such as having nearly no similarities at Layers 2 and 4. These discrepencies may be due to the difference in training data, as it has been shown that SAEs are highly dataset dependent (Kissane et al., 2024b). However, despite these discrepencies in individual features, the feature subspaces we compare appears to be highly similar, which may suggest that while the specific values captured by feature weight vectors may differ, these values lie in similar latent space regions that capture similar semantics.

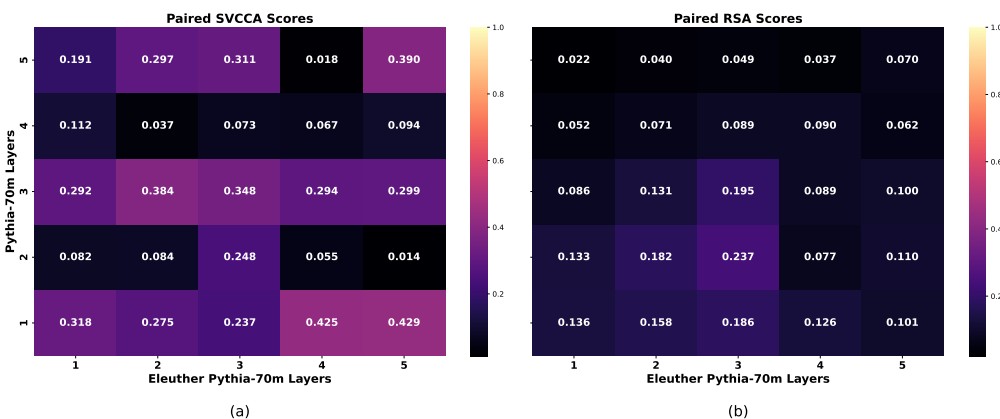

(a)                                              (b)

Figure 12: (a) SVCCA 1-1 paired scores of Eleuther's SAE (trained on 8.2 billion tokens from the Pile) vs our SAE (trained on 100 million tokens from RedPajamas). Both have 32k latents. (b) RSA 1-1 paired scores of the SAEs.

We also train SAEs using 100 million tokens from OpenWebText and compare them to the SAEs we trained on RedPajamas. These results are shown in Figure 13. Similar to the results in Figure 12, we find both similarities and dissimlarities which suggest that SAEs are highly dataset dependent.

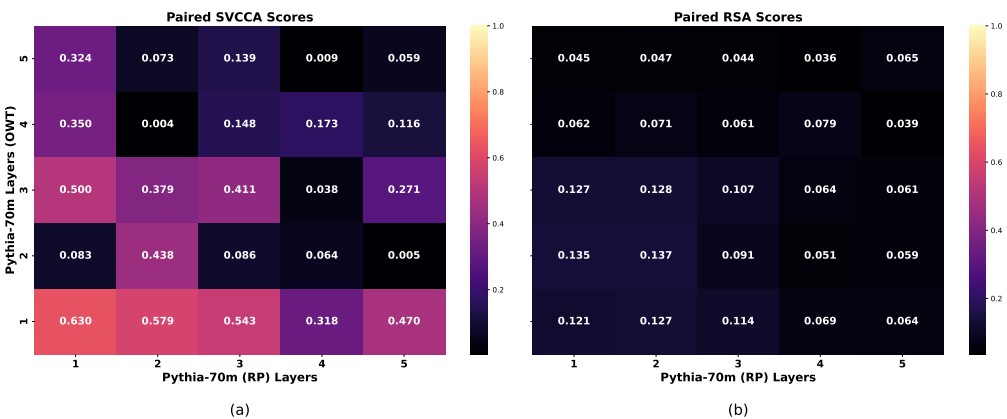

Figure 13: (a) SVCCA 1-1 paired scores of our SAEs trained on RedPajamas (columns, x-axis) vs our SAEs trained on OpenWebText (rows, y-axis). Both have 32k latents. (b) RSA 1-1 paired scores of the SAEs.

## G    CROSS MODEL SAE WIDTH VARIATION RESULTS

We compare what happens to similarity scores as we increase SAE widths for both Pythia-70m and Pythia-160m. All SAEs were trained using 100 million tokens from RedPajamas and use Top-K of 32. Similar to Section §F.3, we do not find noticeable trends in scores as we vary SAE widths. This may be because each SAE varies in the features they capture, so while they capture feature similarities across models, the similar features that they do capture are not the same for each SAE.

Figure 14 SVCCA and RSA 1-1 paired scores of 24k latent SAEs trained on Pythia-160m vs 16k latent SAEs trained on Pythia-70m, and Figure 15 SVCCA and RSA 1-1 paired scores of 49k latent SAEs trained on Pythia-160m vs 32k latent SAEs trained on Pythia-70m. In both figures, while SVCCA scores are moderately high for some layers, the RSA scores are very low.

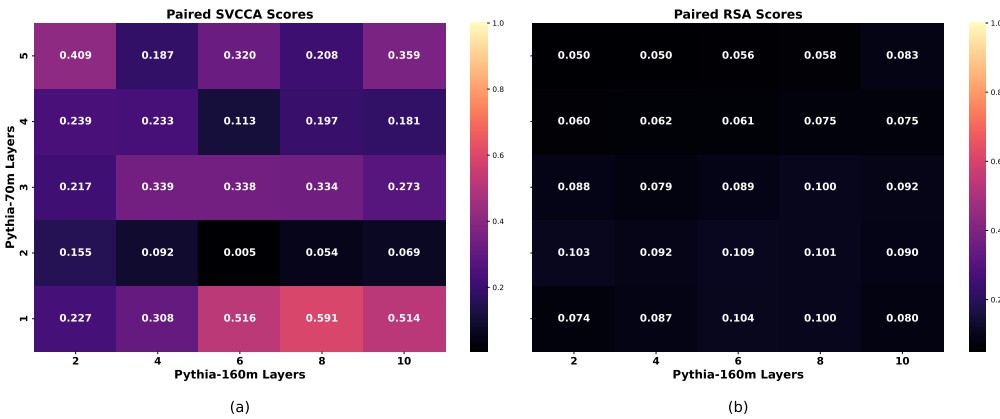

Figure 14: (a) SVCCA 1-1 paired scores of 24k latent SAEs trained on Pythia-160m vs 16k latent SAEs trained on Pythia-70m. (b) RSA 1-1 paired scores of the SAEs.

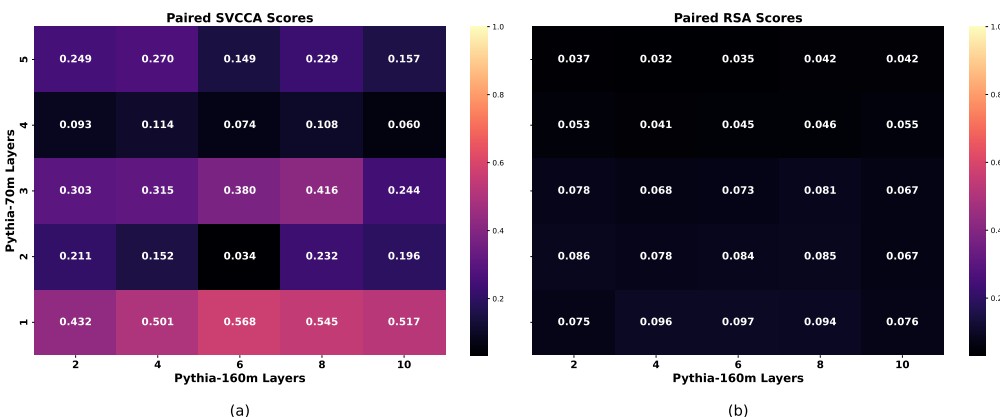

Figure 15: (a) SVCCA 1-1 paired scores of 49k latent SAEs trained on Pythia-160m vs 32k latent SAEs trained on Pythia-70m. (b) RSA 1-1 paired scores of the SAEs.

## H VARYING FACTORS USED TO OBTAIN ACTIVATIONS

### H.1 COMPARING DIFFERENT DATASETS TO OBTAIN ACTIVATIONS

We compare results using different datasets, namely the RedPajamas and OpenWebText datasets, to obtain dataset activations used for matching features by activation correlation. For Eleuther's 32k SAEs trained on Pythia-160m vs on Pythia-70m, Figure 16 displays results from using the RedPajamas dataset, whereas Figure 17 displays results from using the OpenWebText dataset. Overall, the results are similar, with scores from OpenWebText being slightly higher in most cases. This demonstrates that our method is not highly dataset dependent when obtaining activations to pair features by correlation.

Note that these results differ from previous results in Figure 2 because for features matching using Many-to-1 (before filtering), we are using Pythia 160m as the model whose features can have "Many" mappings and Pythia 70m as model whose features can only have "1" mapping, whereas the results in Figure 2 reverse these roles. As implied in Section §L, the results of using a model in the "Many" or "1" places of Many-to-1 are not symmetric, partly due to using max correlation to choose a feature's partner.

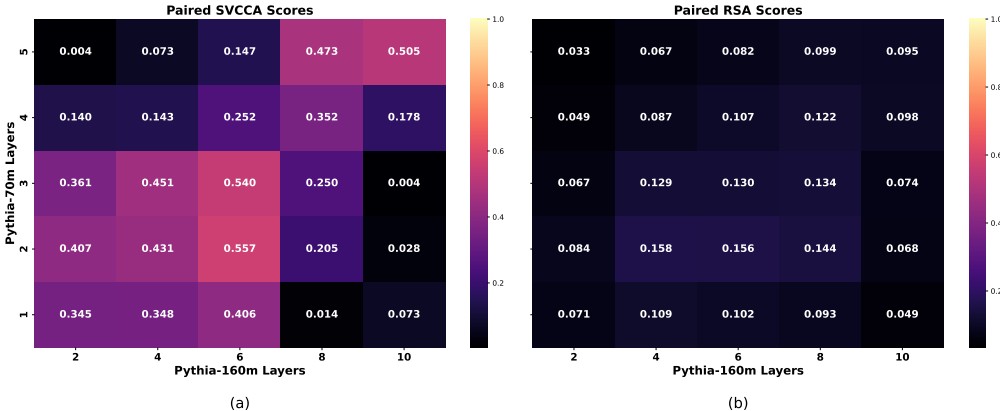

Figure 16: (a) SVCCA 1-1 paired scores of Eleuther's 32k SAEs trained on Pythia-160m vs on Pythia-70m. (b) RSA 1-1 paired scores of the SAEs. The dataset activations were obtained using 40k tokens from the RedPajamas dataset.

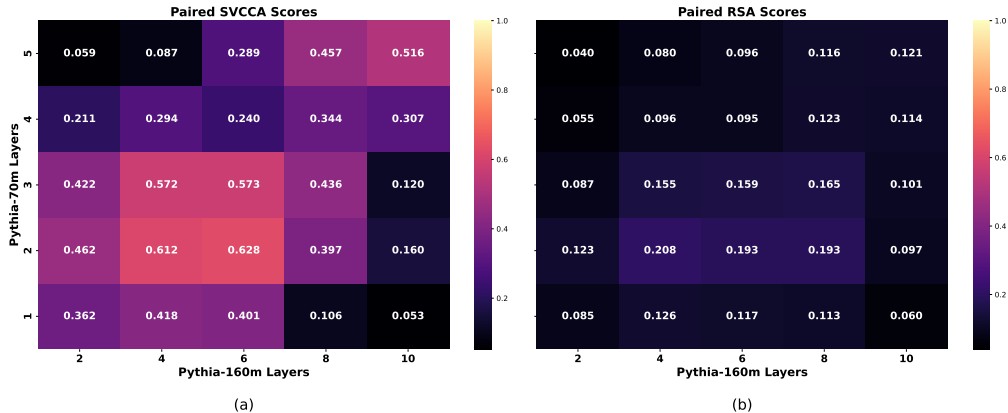

(a)             (b)

Figure 17: (a) SVCCA 1-1 paired scores of Eleuther's 32k SAEs trained on Pythia-160m vs on Pythia-70m. (b) RSA 1-1 paired scores of the SAEs. The dataset activations were obtained using 40k tokens from the OpenWebText dataset.

## H.2 Comparing more data to obtain activations

Figure 18 shows SVCCA Many-1 paired scores of SAEs for layers in Pythia-70m vs layers in Pythia-160m. This is run with 90k tokens, using 300 samples with a max sequence length of 300. Layer 3 appears to be an outlier, perhaps due to errors in feature matching. These results are similar to and consistent with the results of Figure 20, suggesting that other method is not heavily dependent on the number of tokens used to obtain activations.

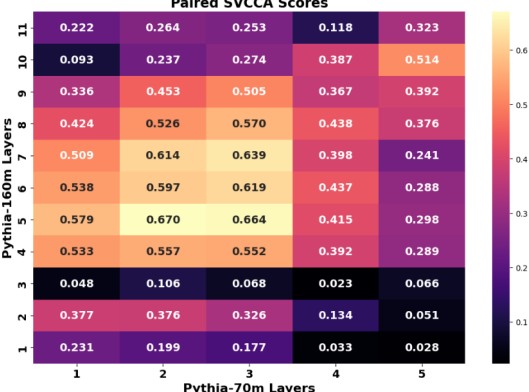

Figure 18: (a) SVCCA Many-1 paired scores of SAEs for layers in Pythia-70m vs layers in Pythia-160m. This is run with 90k tokens, using 300 samples with a max sequence length of 300. Layer 3 appears to be an outlier, perhaps due to errors in feature matching. This is consistent with Figure 20.

## I Using an Analogy to Explain Why this Approach Works

To better describe why this approach works for testing our hypothesis, we can consider the following analogy: given three points $(1, 2, 3)$ on a right triangle, we can think of each point like a feature. One model may label these points as $(A, B, C)$, with each position being a column of a matrix, and another model may label them as $(X, Y, Z)$. However, these points are not ordered the same in each matrix, so we may find them in the order of $(B, C, A)$ and $(Z, X, Y)$. But if we rearrange them into pairs "close enough to" $(1, 2, 3)$ by estimating which points they correspond to on a right triangle, then we can align the two representations, and measure if the relationship among them—a right

triangle—is also similar. This necessitates step 1 in our approach: consistent high scores across models can only be found if these features are paired correctly.

## J UNIVERSAL VS TRUE FEATURES

Our definition of universal features is based on Claim 3 of "Zoom In: An Introduction to Circuits", which defines universality as studying the extent to which analogous features and circuits form across models and tasks (Olah et al., 2020). Previous work has investigated similar claims by showing evidence for highly correlated neurons (Yosinski et al., 2014; Gurnee et al., 2024) and similar feature space representations at hidden layers (Kornblith et al., 2019; Klabunde et al., 2023). Using these definitions of universality, our results find both highly correlated features and similar feature representations in SAEs.

On the other hand, "true features" (Bricken et al., 2023) can be interpreted as atomic linear directions, such that activations are a linear combination of "true feature" directions (Till, 2023), though this definition is still subject to change in future works. This definition for features imposes stricter requirements which do not just rely on high correlation and similar representations, but define "true features" as linear directions that model the same concepts, assuming they exist in ground truth. Previous work has shown that SAEs may capture ground truth features Sharkey et al. (2022), and provided evidence that SAE features are not atomic (Leask et al., 2025), which suggests SAEs may not be sufficient to capture "true features" if they exist.

Chughtai et al. (2023) define universality in terms of *weak* and *strong* universality. Strong universality claims that models trained in similar ways will develop the same features and circuits, whereas weak universality argues that fundamental principles, such as similar functionalities, are learned across models, but their exact implementations may vary considerably. From this perspective, strong universality would correspond more with "true features", while our paper would align closer to measuring weak universality in terms of similar feature space correlations.

**The Importance of Analogous Features.** While features across models may not capture the same ground truth feature representations, having analogous feature spaces could suggest that there may be transformations or mappings between the representation spaces of models which reveal their similarity. As such, this has implications for transfer learning and model stitching in which representations, including cross-layer circuits, from one model may be transferred to another (possibly larger) model through a mapping model (Bansal et al., 2021), allowing for feature functionality transfer. These studies may also have implications for furthering studies in neuroscience, as past research has been conducted on measuring the extent of alignment across biological and artificial neural networks (Goldstein et al., 2024), and previous work has also found high-low feature detectors in both vision models (Schubert et al., 2021) and mouse brains (Ding et al., 2023).

Currently, there are no standardized definitions for *analogous features*, though it may be possible to develop them. For instance, one possible way to better formalize a definition of analogous features would be along the lines of using a criteria akin to $(f \circ g)(x) = (h \circ f)(y)$, given that $f$ is a mapping, while $g$ and $h$ are relations, such as distances, between points $x \in X$ and $y \in Y$ for spaces $X$ and $Y$.

This paper seeks to measure the extent of universality in terms of "analogous features in representational space", and unlike Huh et al. (2024), does not claim that models converge towards finding universal features. This follows previous papers which also measured the extent to which different neural networks learn the same representation (Wang et al., 2018a). We provide evidence that shows there are non-negligible signals of universality captured by subsets of SAE feature spaces across models, but also claim that many feature space subsets in SAEs are dissimilar in terms of their representations and relations in feature space. Still, since this may be due to the limitations of SAEs in capturing features (Engels et al., 2024), these results do not conclude that in these dissimilar areas, models do not share universal features, and further work must be done in this area to draw stronger conclusions.

Stated another way, this paper is not claiming that different SAEs consistently learn the same universal features captured by LLMs. Rather, it is claiming that LLMs learn weakly universal features, and that SAEs can capture not only these features, but their relations. However, one notable limitation is that SAEs cannot capture these features consistently.

## K    ILLUSTRATED STEPS OF COMPARISON METHOD

Figure 19 demonstrates steps 1 to 3 to carry out our similarity comparison experiments. Since SVCCA and RSA are carried out by using the features (or data samples) on the rows, we take the transpose of the decoder weight matrices, which represent the features on its columns.

Note that while these measures are permutation-invariant in their columns, they are not invariant in regards to their rows, which must still be paired correctly (Klabunde et al., 2023). Additionally, these measures require both matrices to have the same number of rows. For SAEs with the same number of feature columns, this is not an issue, but for SAEs with different number of columns, we only consider 1-1 mappings.

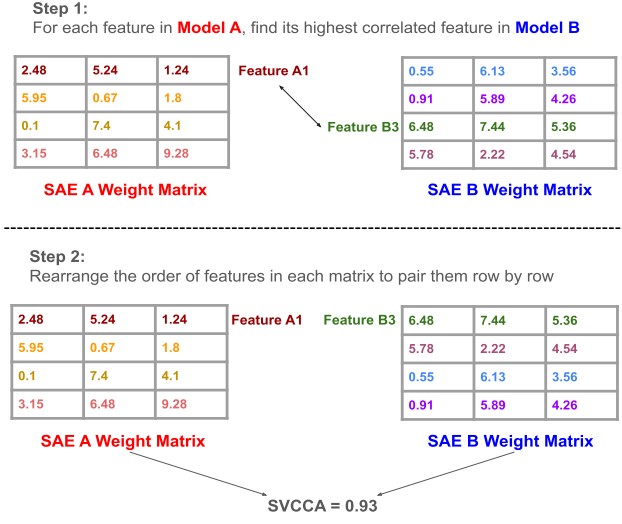

Figure 19: Steps to Main Results. We first find correlated pairs to solve neuron permutation, and then apply similarity metrics to solve latent space rotation issues.

## L    NOISE OF MANY-TO-1 MAPPINGS

Overall, we do not aim for mappings that uniquely pair every feature of both spaces as we hypothesize that it is unlikely that all the features are the same in each SAE; rather, we are looking for large subspaces where the features match by activations and we can map one subspace to another.

We hypothesize that these many-to-1 features may contribute a lot of noise to feature space alignment similarity, as information from a larger space (eg. a ball) is collapsed into a dimension with much less information (eg. a point). Thus, when we only include 1-1 features in our similarity scores, we receive scores with much less noise.

As shown in the comparisons of Figure 20 to Figure 21, we find the 1-1 feature mappings give slightly higher scores for many layer pairs, like for the SVCCA scores at L2 vs L3 for Pythia-70m vs Pythia-160m, though they give slightly lower scores for a few layer pairs, like for the SVCCA scores at L5 vs L4 for Pythia-70m vs Pythia-160m. Given that most layer pair scores are slightly higher, we use 1-1 feature mappings in our main text results.

Additionally, we notice that for semantic subspace experiments, 1-1 gave vastly better scores than many-to-1. We hypothesize that this is because since these feature subspaces are smaller, the noise from the many-to-1 mappings have a greater impact on the score.

We also looked into using more types of 1-1 feature matching, such as pairing many-to-1 features with their "next highest" or using efficient methods to first select the mapping pair with the highest

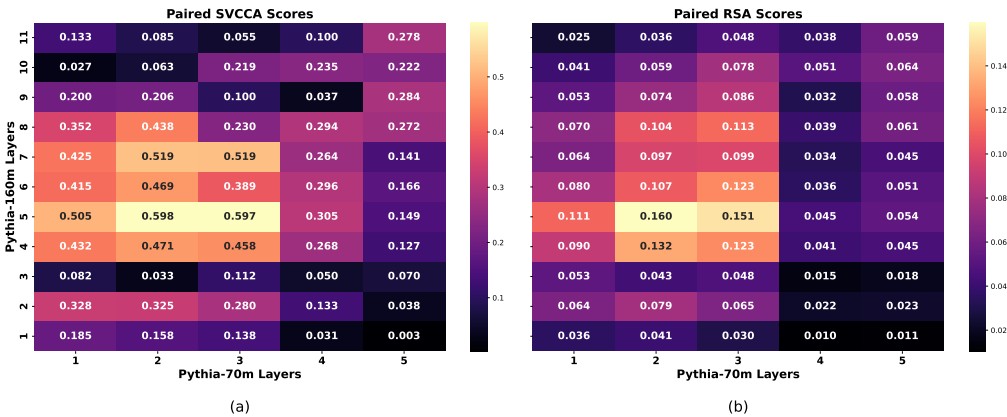

Figure 20: (a) SVCCA and (b) RSA **Many-to-1** paired scores of SAEs for layers in Pythia-70m vs layers in Pythia-160m. We note there appears to be an "anomaly" at L2 vs L3 with a low SVCCA score; we find that taking 1-1 features greatly increases this score from 0.03 to 0.35, as shown and explained in Figure 2.

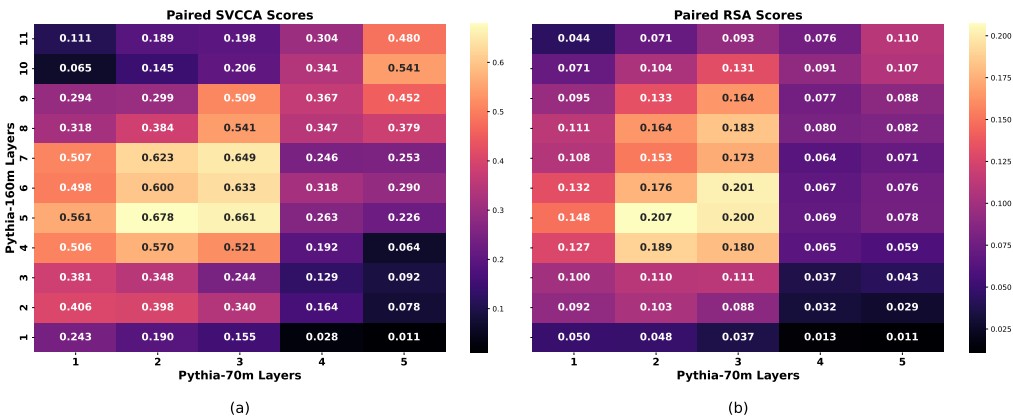

Figure 21: (a) SVCCA and (b) RSA **1-1** paired scores of SAEs for layers in Pythia-70m vs layers in Pythia-160m. Compared to Figure 20, some of the scores are slightly higher, and the SVCCA score at L2 vs L3 for 70m vs 160m is much higher. On the other hand, some scores are slightly lower, such as SVCCA at L5 vs L4 for 70m vs 160m. This is the same figure as Figure 2; it is shown here again for easier comparison to the Many-to-1 scores in Figure 20.

correlation, taking those features off as candidates for future mappings, and continuing this for the next highest correlations. This also appeared to work well, though further investigation is needed. More analysis can also be done for mapping one feature from SAE A to many features in SAE B.

## M    NOISE OF NON-CONCEPT FEATURE PAIRINGS

We define "non-concept features" as features which are not modeling specific concepts that can be mapped well across models. Their highest activations are on new lines, spaces, punctuation, and EOS tokens. As such, they may introduce noise when computing similarity scores, as their removal appears to greatly improve similarity scores. We hypothesize that one reason they introduce noise is that these could be tokens that aren't specific to concepts, but multiple concepts. More specifically, since tokens are context dependent to a model, as information propagates through layers, the activations at a token position can aggregate information about that position's neighbors. Therefore, it is possible that these features are not "noisy" features, but are capturing concepts which are not represented well purely by

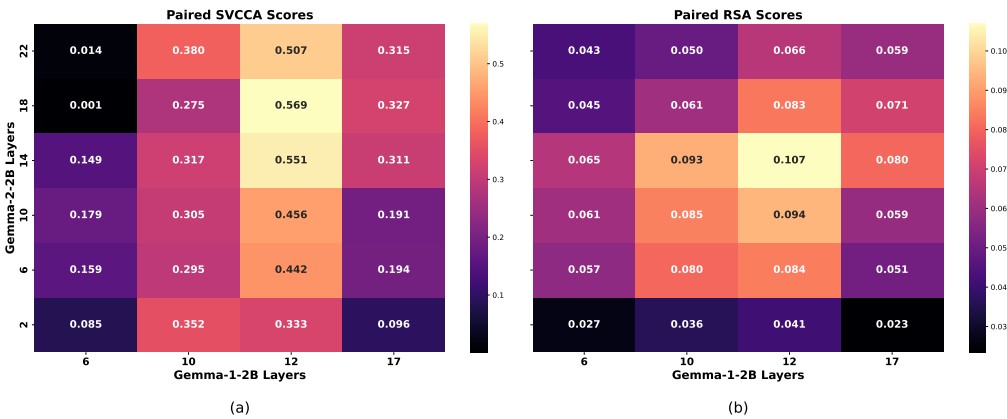

Figure 22: (a) SVCCA and (b) RSA **Many-to-1** paired scores of SAEs for layers in Gemma-1-2B vs layers in Gemma-2-2B. We obtain similar scores compared to the 1-1 paired scores in Figure 3.

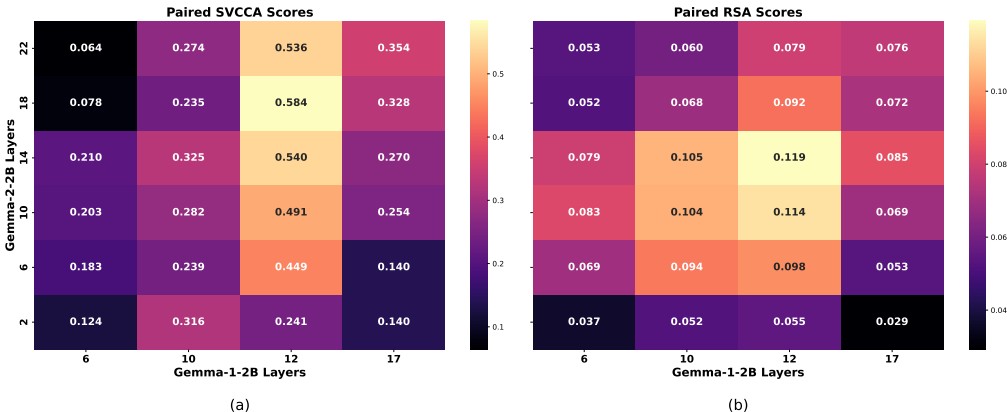

Figure 23: (a) SVCCA and (b) RSA **1-1** paired scores of SAEs for layers in Gemma-1-2B vs layers in Gemma-2-2B. Compared to Figure 22, some of the scores are slightly higher, whiel some are slightly lower. This is the same figure as Figure 3; it is shown here again for easier comparison to the Many-to-1 scores in Figure 20.

activation correlation based on highest activating positions, and thus they are not well matched via activation correlation.

For instance, say feature $X$ from model $A$ fires highly on "!" in the sample "I am BOB!", while feature $Y$ from model $B$ fires highly on "!" in the sample, "that cat ate!"

A feature $X$ that activates on "!" in "BOB!" may be firing for names, while a feature $Y$ that activates on "!" in "ate!" may fire on food-related tokens. As such, features activating on non-concept features are not concept specific, and we could map a feature firing on names to a feature firing on food. We call these possibly "incoherent" mappings "non-concept mappings". By removing non-concept mappings, we show that SAEs reveal highly similar feature spaces.

A second hypothesis is that there are not diverse enough tokens to allow features to fire on what they should fire on. As such, providing more data to obtain correlations may allow these features to activate on more concepts, and thus be matched more accurately.

We filter "non-concept features" that fire, for at least one of their top 5 dataset examples, on a "non-concept" keyword. The list of "non-concept" keywords we use is given below:

- \\n
- \n

- (empty string)
- (space)
- .
- ,
- !
- ?
- -
- respective padding token of tokenizer (`<bos>`, `<|endoftext|>`, etc.)

There may exist other non-concept features that we did not filter, such as colons, brackets and carats. We do not include arithmetic symbols, as those are domain specific to specific concepts. Additionally, we have yet to perform a detailed analysis on which non-concept features influence the similarity scores more than others; this may be done in future work.

## N   CONCEPT CATEGORIES LIST AND FURTHER SEMANTIC SUBSPACE ANALYSIS

The keywords used for each concept group for the semantic subspace experiments are given in Table 4. We generate keywords using GPT-4 (Bubeck et al., 2023) using the prompt "give a list of N single token keywords as a python list that are related to the concept of C", such that N is a number (we use N = 50 or 100) and C is a concept like Emotions. We then manually filter these lists to avoid using keywords which have meanings outside of the concept group (eg. "even" does not largely mean "divisible by two" because it is often used to mean "even if..."). We also aim to select case-insensitive keywords which are single-tokens with no spaces. [3]

When searching for keyword matches from each feature's list of top 5 highest activating tokens, we use keyword matches that avoid features with dataset samples which use the keywords in compound words. For instance, when searching for features that activate on the token "king" from the People/Roles concept, we avoid using features that activate on "king" when "king" is a part of the word "seeking", because "seeking" is unrelated to People/Roles.

We perform interpretability experiments to check, of the features kept, which keywords are activated on. We find that many feature pairs activate on the same keyword and are monosemantic. Most keywords in each concept group are not kept. Additionally, for the layer pairs we checked on, after filtering there were only a few keywords that multiplied features fired on. This shows that the high similarity is not because the same keyword is over-represented in a space.

For instance, for Layer 3 in Pythia-70m vs Layer 5 in Pythia-160m, and for the concept category of Emotions, we find which keywords from the category appear in the top 5 activating tokens of each of the Pythia-70m features from the 24 feature mappings described in Table 1. We only count a feature if it appears once in the top 5 of a feature's top activating tokens; if three out of five of a feature's top activating tokens are "smile", then smile is only counted once. The results for Models A and B are given in Table 3. Not only are their counts similar, indicating similar features, but there is not a great over-representation of features. There are a total of 24 features, and a total of 25 keywords (as some features can activate on multiple keywords in their top 5). We note that even if a feature fires for a keyword, that does not mean that that feature's purpose is only to "recognize that keyword", as dataset examples as just one indicator of what a feature's purpose is (which still can have multiple, hard-to-interpret purposes).

However, as there are many feature pairs, we did not perform a thorough analysis of this yet, and thus did not include this analysis in this paper. We also check this in LLMs, and find that the majority of LLM neurons are polysemantic. We do not just check the top 5 dataset samples for LLM neurons, but the top 90, as the SAEs have an expansion factor that is around 16x or 32x bigger than the LLM dimensions they take in as input.

Calendar is a subset of Time, removing keywords like "after" and "soon", and keeping only "day" and "month" type of keywords pertaining to dates.

---

[3]However, not all the tokens in the Table 4 are single token.

Table 3: Count of Model A vs Model B features with keywords from the Emotion category in the semantic subspace found in Table 1.

| Model A | | | Model B | |
|---|---|---|---|---|
| **Keyword** | **Count** | | **Keyword** | **Count** |
| free | 4 | | free | 5 |
| pain | 4 | | pain | 4 |
| smile | 3 | | love | 3 |
| love | 2 | | hate | 2 |
| hate | 2 | | calm | 2 |
| calm | 2 | | smile | 2 |
| sad | 2 | | kind | 1 |
| kind | 1 | | shy | 1 |
| shy | 1 | | doubt | 1 |
| doubt | 1 | | trust | 1 |
| trust | 1 | | peace | 1 |
| peace | 1 | | joy | 1 |
| joy | 1 | | sad | 1 |

Future work can test on several groups of unrelated keywords, manually inspecting to check that each word belongs to its own category. Another test to run is to compare the score of a semantic subspace mapping to a mean score of subspace mapping of the same number of correlated features, using the correlation matrix of the entire space. Showing that these tests result in high p-values will provide more evidence that selecting any subset of correlated features is not enough, and that having the features be semantically correlated with one another yields higher similarity.

## O  ADDITIONAL RESULTS FOR PYTHIA AND GEMMA

As shown in Figure 24 for Pythia and Figure 25 for Gemma, we also find that mean activation correlation does not always correlate with the global similarity metric scores. For instance, a subset of feature pairings with a moderately high mean activation correlation (eg. 0.6) may yield a low SVCCA score (eg. 0.03).

The number of features after filtering non-concept features and many-to-1 features are given in Tables 5 and 6. The number of features after filtering non-concept features and many-to-1 features are given in Tables 7 and 8.

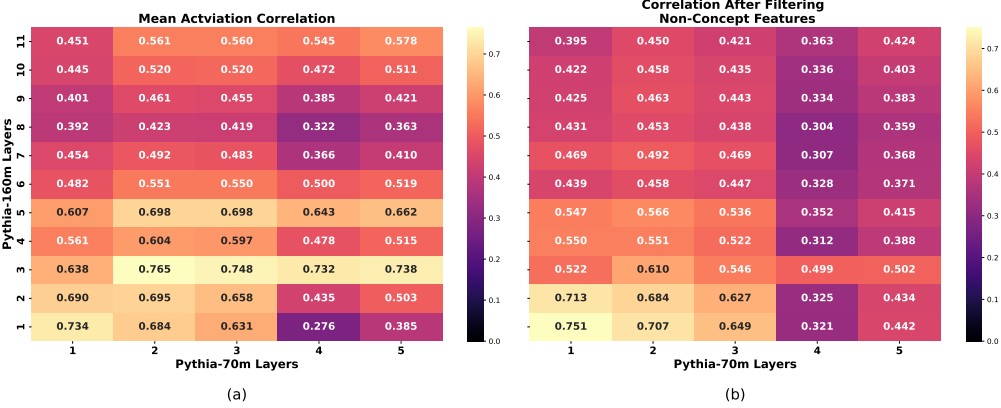

Figure 24: Mean Activation Correlation before (a) and after (b) filtering non-concept features for Pythia-70m vs Pythia-160m. We note these patterns are different from those of the SVCCA and RSA scores in Figure 20, indicating that these three metrics each reveal different patterns not shown by other metrics.

Table 4: Keywords Organized by Category

| Category | Keywords |
| --- | --- |
| Time | day, night, week, month, year, hour, minute, second, now, soon, later, early, late, morning, evening, noon, midnight, dawn, dusk, past, present, future, before, after, yesterday, today, tomorrow, next, previous, instant, era, age, decade, century, millennium, moment, pause, wait, begin, start, end, finish, stop, continue, forever, constant, frequent, occasion, season, spring, summer, autumn, fall, winter, anniversary, deadline, schedule, calendar, clock, duration, interval, epoch, generation, period, cycle, timespan, shift, quarter, term, phase, lifetime, timeline, delay, prompt, timely, recurrent, daily, weekly, monthly, yearly, annual, biweekly, timeframe |
| Calendar | day, night, week, month, year, hour, minute, second, morning, evening, noon, midnight, dawn, dusk, yesterday, today, tomorrow, decade, century, millennium, season, spring, summer, autumn, fall, winter, calendar, clock, daily, weekly, monthly, yearly, annual, biweekly, timeframe |
| People/Roles | man, girl, boy, kid, dad, mom, son, sis, bro, chief, priest, king, queen, duke, lord, friend, clerk, coach, nurse, doc, maid, clown, guest, peer, punk, nerd, jock |
| Nature | tree, grass, stone, rock, cliff, hill, dirt, sand, mud, wind, storm, rain, cloud, sun, moon, leaf, branch, twig, root, bark, seed, tide, lake, pond, creek, sea, wood, field, shore, snow, ice, flame, fire, fog, dew, hail, sky, earth, glade, cave, peak, ridge, dust, air, mist, heat |
| Emotions | joy, glee, pride, grief, fear, hope, love, hate, pain, shame, bliss, rage, calm, shock, dread, guilt, peace, trust, scorn, doubt, hurt, wrath, laugh, cry, smile, frown, gasp, blush, sigh, grin, woe, spite, envy, glow, thrill, mirth, bored, cheer, charm, grace, shy, brave, proud, glad, mad, sad, tense, free, kind |
| MonthNames | January, February, March, April, May, June, July, August, September, October, November, December |
| Countries | USA, Canada, Brazil, Mexico, Germany, France, Italy, Spain, UK, Australia, China, Japan, India, Russia, Korea, Argentina, Egypt, Iran, Turkey |
| Biology | gene, DNA, RNA, virus, bacteria, fungus, brain, lung, blood, bone, skin, muscle, nerve, vein, organ, evolve, enzyme, protein, lipid, membrane, antibody, antigen, ligand, substrate, receptor, cell, chromosome, nucleus, cytoplasm |

Table 5: Number of Features in each Layer Pair Mapping after filtering Non-Concept Features for Pythia-70m (cols) vs Pythia-160m (rows) out of a total of 32768 SAE features in both models.

| Layer | 1 | 2 | 3 | 4 | 5 |
| --- | --- | --- | --- | --- | --- |
| 1 | 23600 | 21423 | 26578 | 19696 | 19549 |
| 2 | 16079 | 14201 | 17578 | 12831 | 12738 |
| 3 | 7482 | 7788 | 7841 | 7017 | 6625 |
| 4 | 12756 | 11195 | 13349 | 9019 | 9357 |
| 5 | 7987 | 6825 | 8170 | 5367 | 5624 |
| 6 | 10971 | 9578 | 10937 | 8099 | 8273 |
| 7 | 15074 | 12988 | 16326 | 12720 | 12841 |
| 8 | 14445 | 12580 | 15300 | 11779 | 11942 |
| 9 | 13320 | 11950 | 14338 | 11380 | 11436 |
| 10 | 9834 | 8573 | 9742 | 7858 | 8084 |
| 11 | 6936 | 6551 | 7128 | 5531 | 6037 |

Table 6: Number of **1-1** Features in each Layer Pair Mapping after filtering Non-Concept Features for Pythia-70m (cols) vs Pythia-160m (rows) out of a total of 32768 SAE features in both models.

| Layer | 1 | 2 | 3 | 4 | 5 |
|---|---|---|---|---|---|
| 1 | 7553 | 5049 | 5853 | 3244 | 3115 |
| 2 | 6987 | 4935 | 5663 | 3217 | 3066 |
| 3 | 4025 | 3152 | 3225 | 2178 | 1880 |
| 4 | 6649 | 5058 | 6015 | 3202 | 3182 |
| 5 | 5051 | 4057 | 4796 | 2539 | 2596 |
| 6 | 5726 | 4528 | 5230 | 3226 | 3031 |
| 7 | 7017 | 5659 | 6762 | 4032 | 3846 |
| 8 | 6667 | 5179 | 6321 | 3887 | 3808 |
| 9 | 6205 | 4820 | 5773 | 3675 | 3682 |
| 10 | 4993 | 3859 | 4602 | 3058 | 3356 |
| 11 | 3609 | 2837 | 3377 | 2339 | 2691 |

Table 7: Number of Features in each Layer Pair Mapping after filtering Non-Concept Features for Gemma-1-2B (cols) vs Gemma-2-2B (rows) out of a total of 16384 SAE features in both models.

| Layer | 6 | 10 | 12 | 17 |
|---|---|---|---|---|
| 2 | 8926 | 8685 | 8647 | 4816 |
| 6 | 4252 | 4183 | 4131 | 2474 |
| 10 | 6458 | 6320 | 6277 | 3658 |
| 14 | 4213 | 4208 | 4214 | 2672 |
| 18 | 3672 | 3679 | 3771 | 2515 |
| 22 | 4130 | 4100 | 4168 | 3338 |

Table 8: Number of **1-1** Features in each Layer Pair Mapping after filtering Non-Concept Features for Gemma-1-2B (cols) vs Gemma-2-2B (rows) out of a total of 16384 SAE features in both models.

| Layer | 6 | 10 | 12 | 17 |
|---|---|---|---|---|
| 2 | 3427 | 3874 | 3829 | 1448 |
| 6 | 3056 | 3336 | 3261 | 1266 |
| 10 | 4110 | 4650 | 4641 | 1616 |
| 14 | 2967 | 3524 | 3553 | 1414 |
| 18 | 2636 | 3058 | 3239 | 1543 |
| 22 | 2646 | 3037 | 3228 | 1883 |

Table 9: RSA scores and random mean results for 1-1 semantic subspaces of L3 of Pythia-70m vs L5 of Pythia-160m. P-values are taken for 1000 samples in the null distribution.

| Concept Subspace | Number of 1-1 Features | Paired Mean | Random Shuffling Mean | p-value |
|---|---|---|---|---|
| Time | 228 | 0.10 | 0.00 | 0.00 |
| Calendar | 126 | 0.09 | 0.00 | 0.00 |
| Nature | 46 | 0.22 | 0.00 | 0.00 |
| MonthNames | 32 | 0.76 | 0.00 | 0.00 |
| Countries | 32 | 0.10 | 0.00 | 0.03 |
| People/Roles | 31 | 0.18 | 0.00 | 0.01 |
| Emotions | 24 | 0.46 | 0.00 | 0.00 |

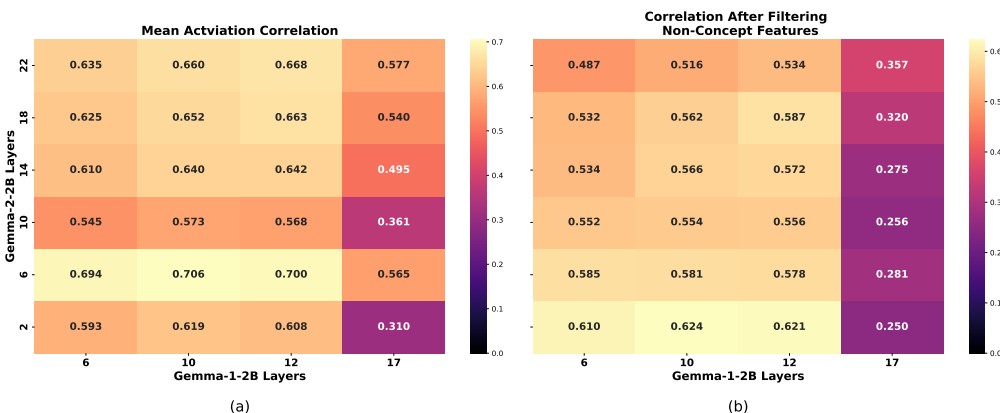

Figure 25: Mean Activation Correlation before (a) and after (b) filtering non-concept features for Gemma-1-2B vs Gemma-2-2B. We note these patterns are different from those of the SVCCA and RSA scores in Figure 22, indicating that these three metrics each reveal different patterns not shown by other metrics.

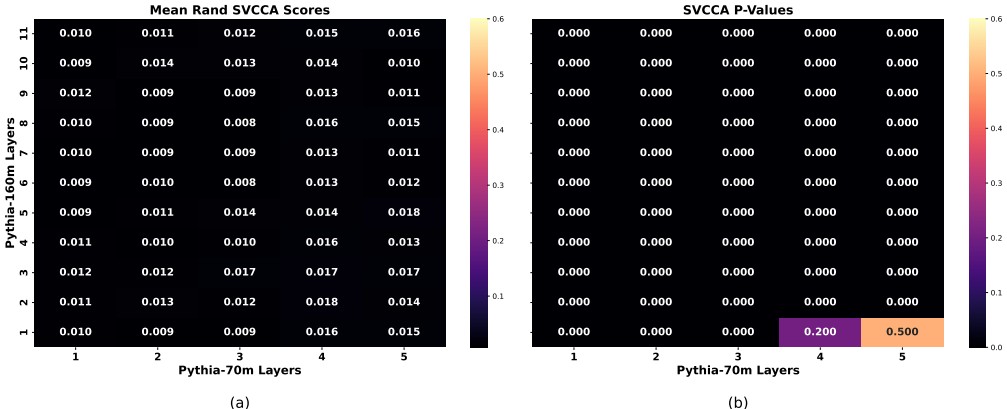

Figure 26: Mean Randomly Paired SVCCA scores and P-values of SAEs for layers in Pythia-70m vs Pythia-160m. Compared to Paired Scores in Figure 20, these are all very low.

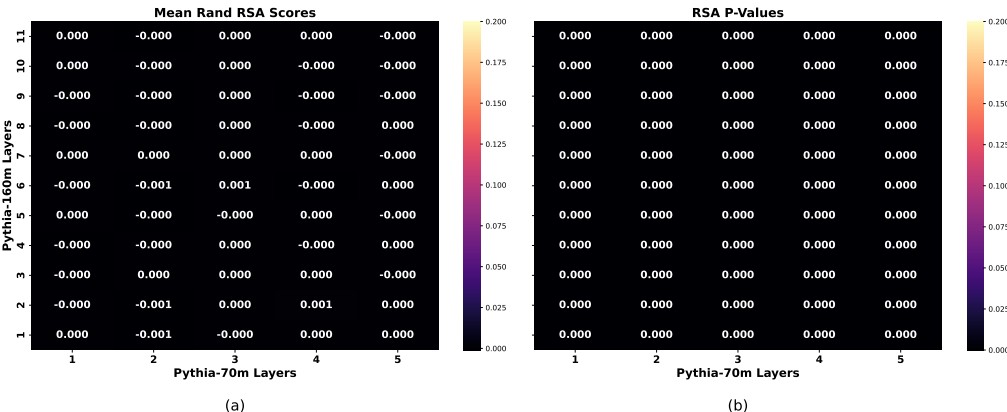

Figure 27: Mean Randomly Paired RSA scores and P-values of SAEs for layers in Pythia-70m vs Pythia-160m. Compared to Paired Scores in Figure 20, these are all very low.

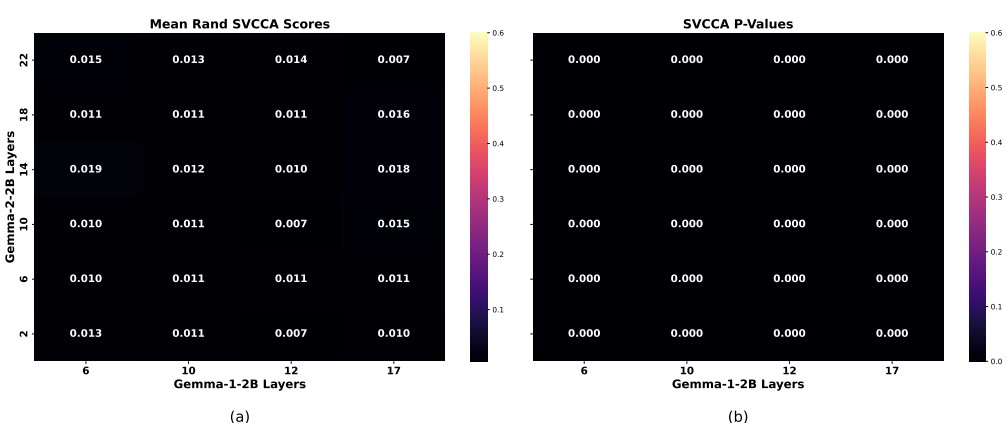

Figure 28: Mean Randomly Paired SVCCA scores and P-values of SAEs for layers in Gemma-1-2B vs layers in Gemma-2-2B. Compared to Paired Scores in Figure 22, these are all very low.

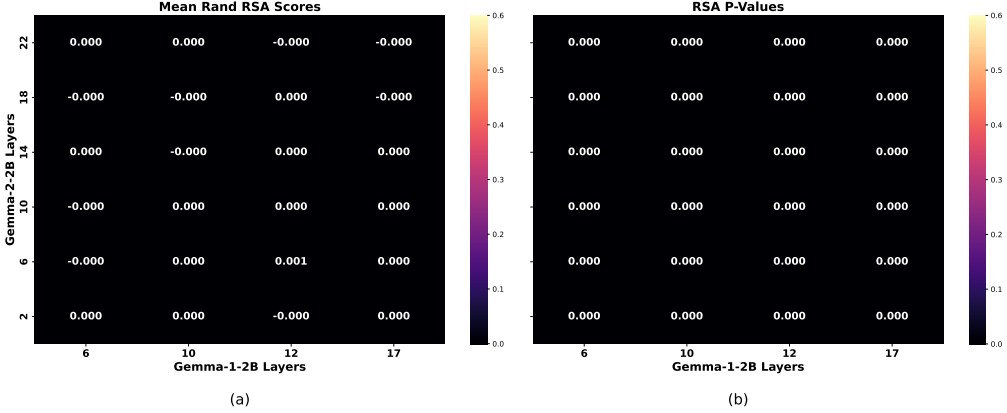

Figure 29: Mean Randomly Paired RSA scores and P-values of SAEs for layers in Gemma-1-2B vs layers in Gemma-2-2B. Compared to Paired Scores in Figure 22, these are all very low.

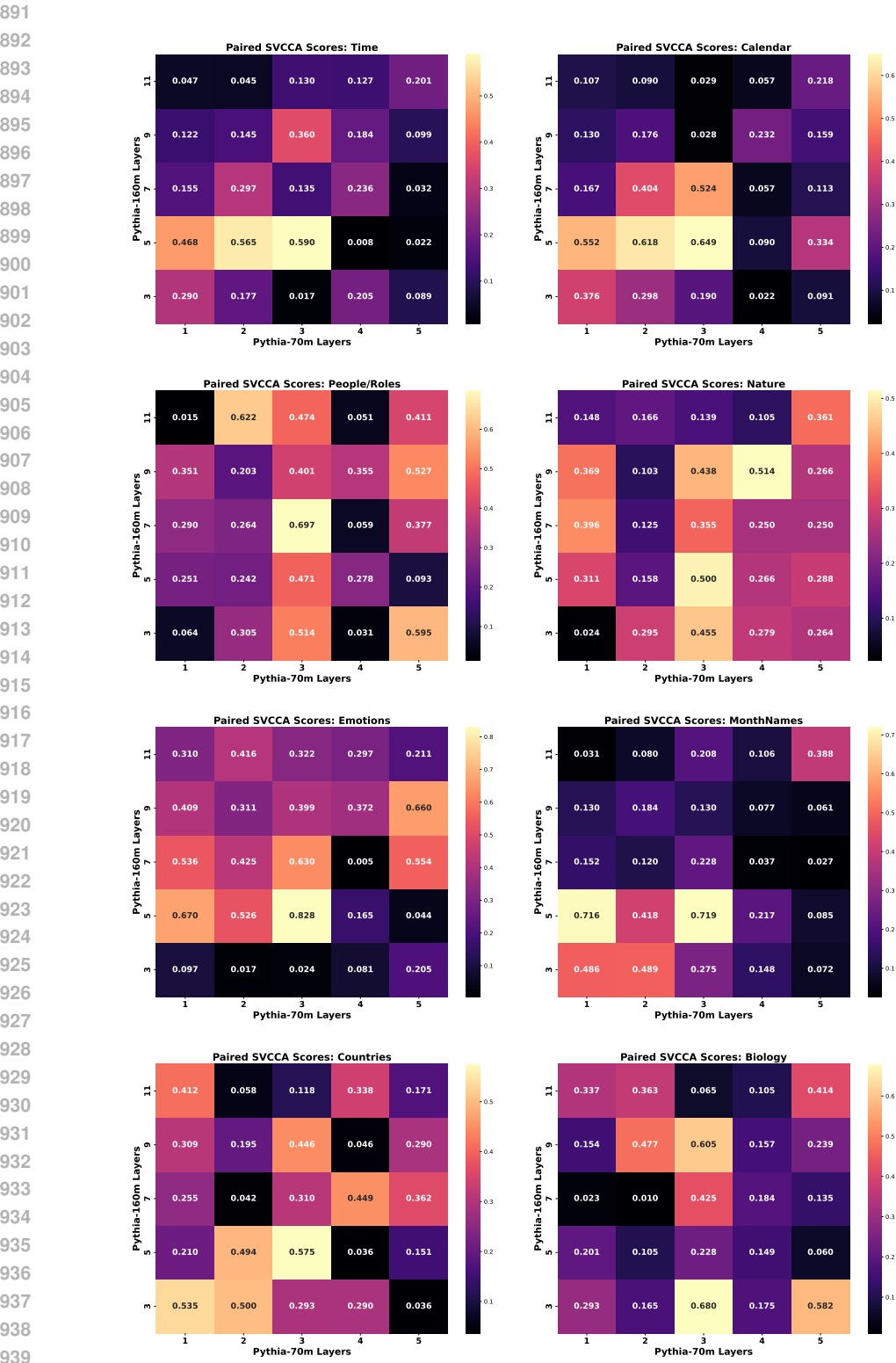

Figure 30: 1-1 Paired SVCCA scores of SAEs for layers in Pythia-70m vs layers in Pythia-160m for Concept Categories. Middle layers appear to be the most similar with one another.

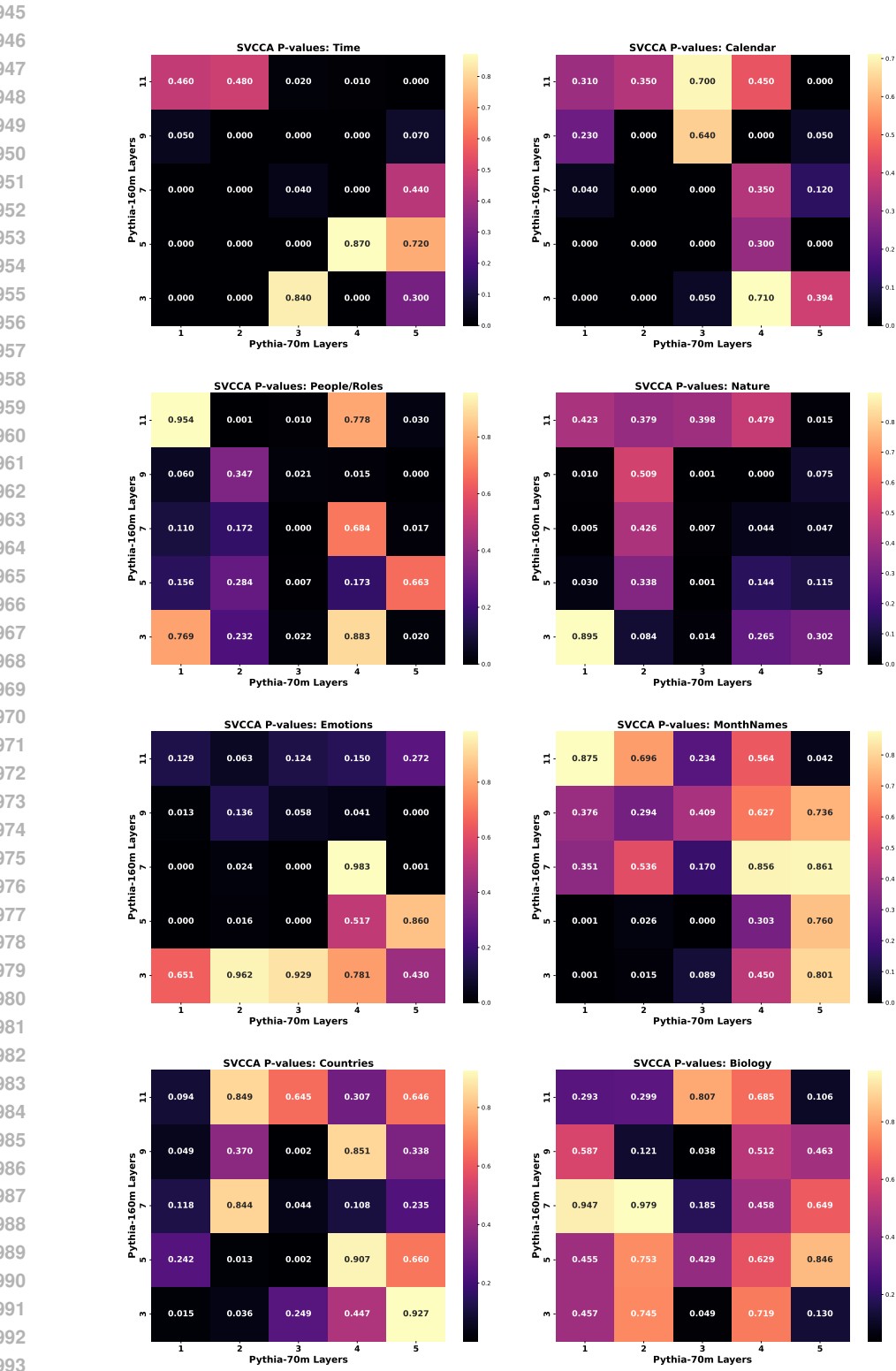

Figure 31: 1-1 SVCCA p-values of SAEs for layers in Pythia-70m vs layers in Pythia-160m for Concept Categories. A lower p-value indicates more statistically meaningful similarity.

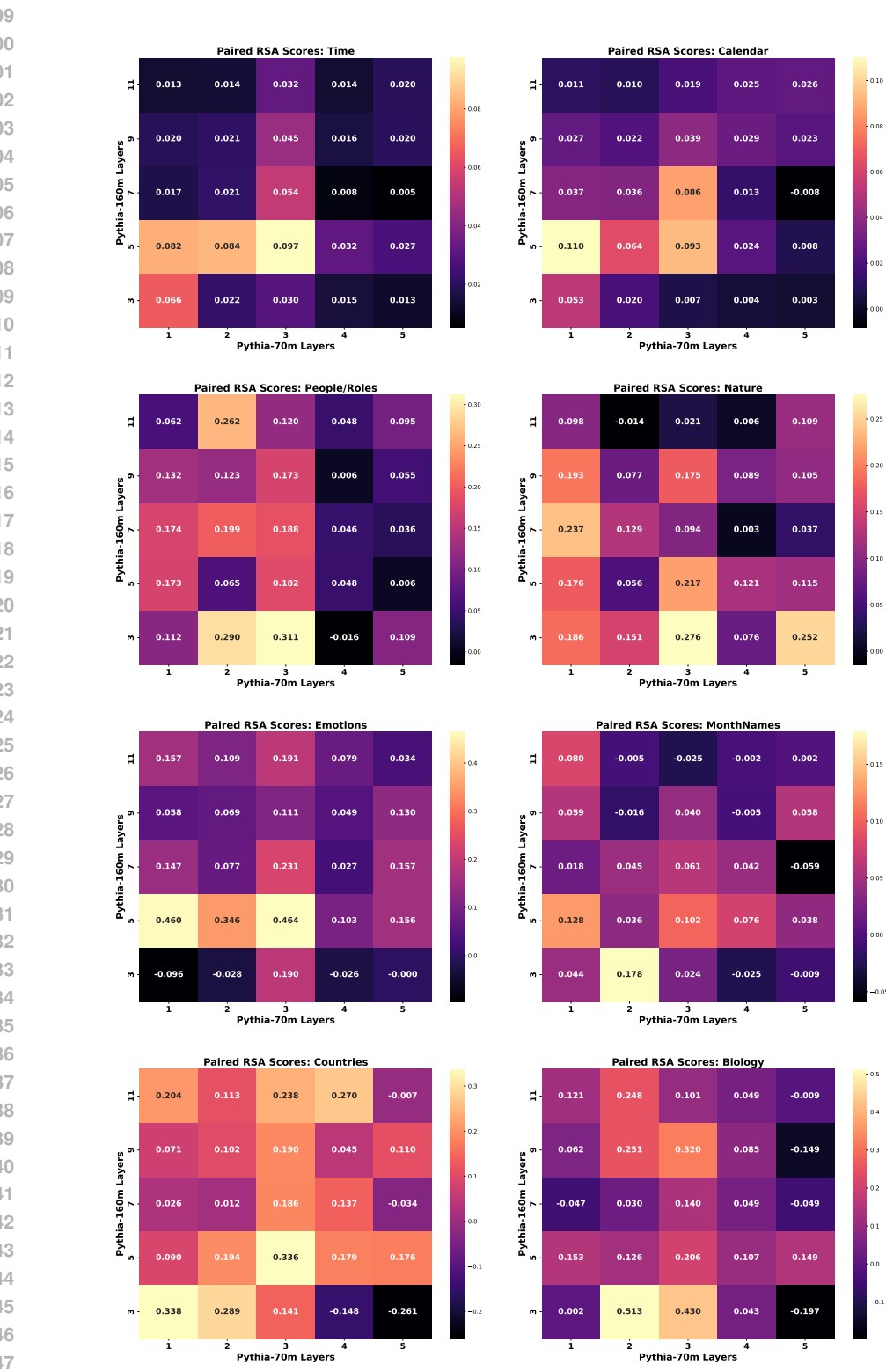

Figure 32: 1-1 Paired RSA scores of SAEs for layers in Pythia-70m vs layers in Pythia-160m for Concept Categories. Middle layers appear to be the most similar with one another.

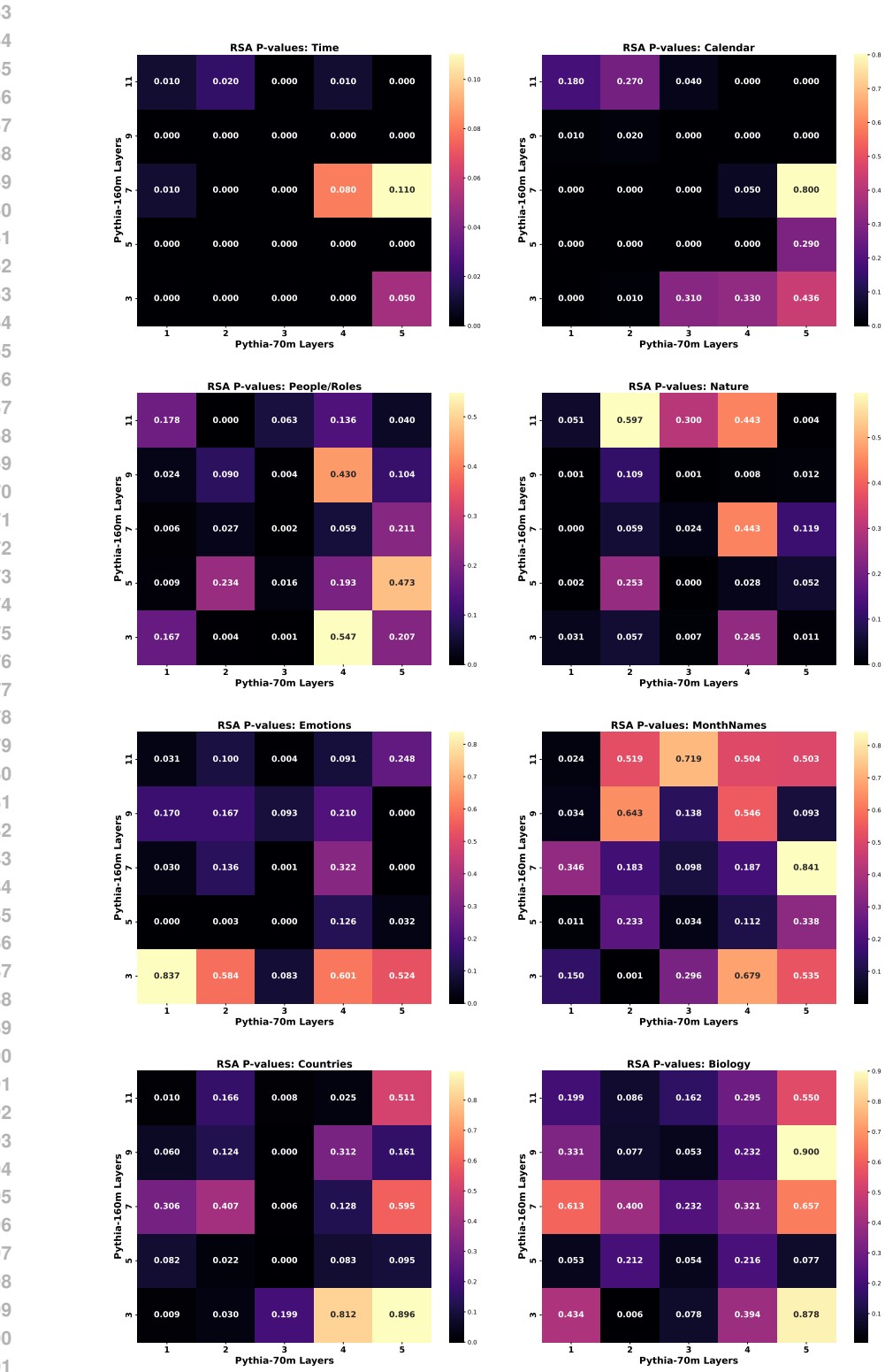

Figure 33: 1-1 RSA p-values of SAEs for layers in Pythia-70m vs layers in Pythia-160m for Concept Categories. A lower p-value indicates more statistically meaningful similarity.

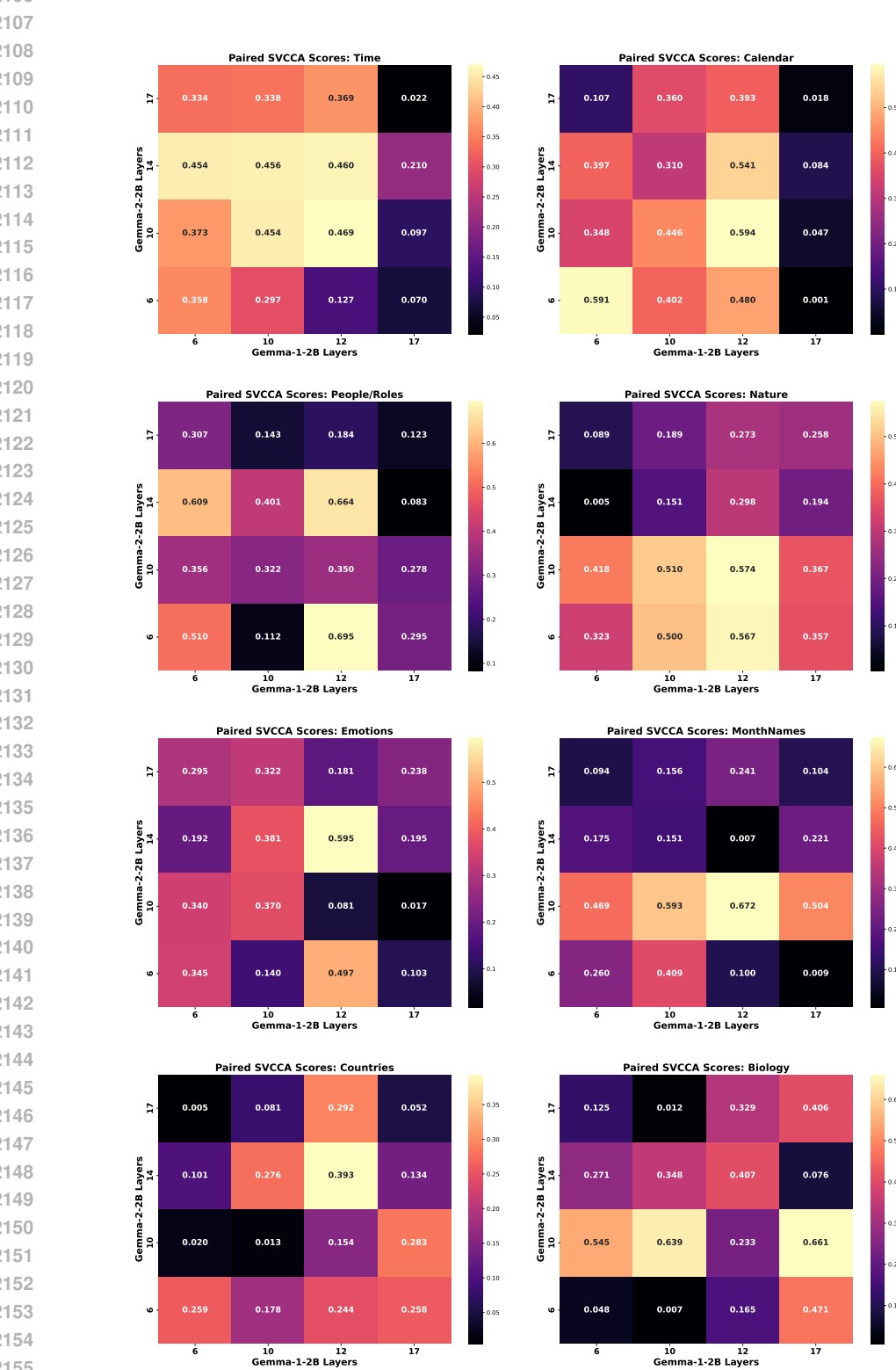

Figure 34: 1-1 Paired SVCCA scores of SAEs for layers in Gemma-1-2b vs layers in Gemma-2-2b for Concept Categories. Middle layers appear to be the most similar with one another.

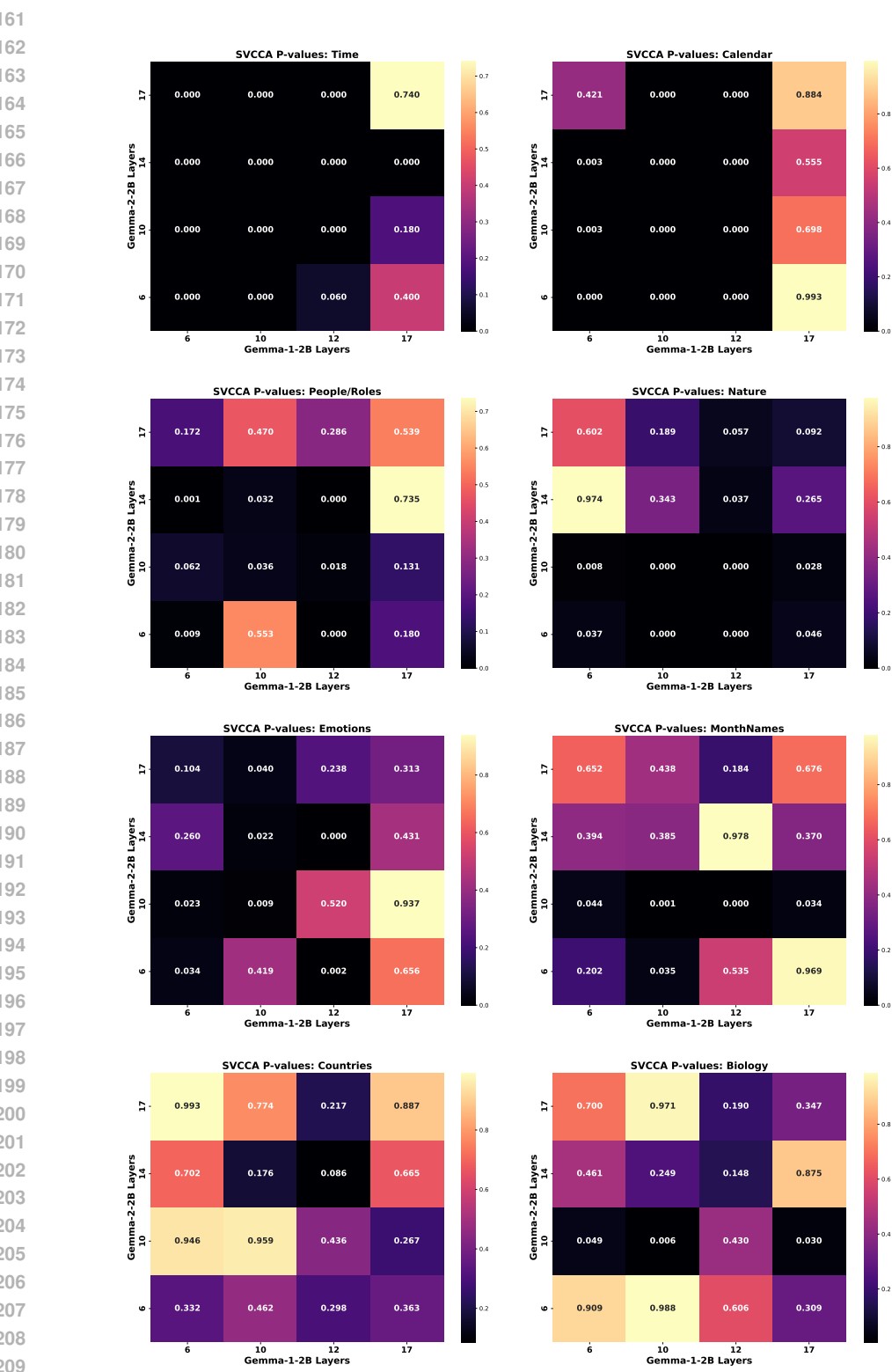

Figure 35: 1-1 SVCCA p-values of SAEs for layers in Gemma-1-2b vs layers in Gemma-2-2b for Concept Categories. A lower p-value indicates more statistically meaningful similarity.

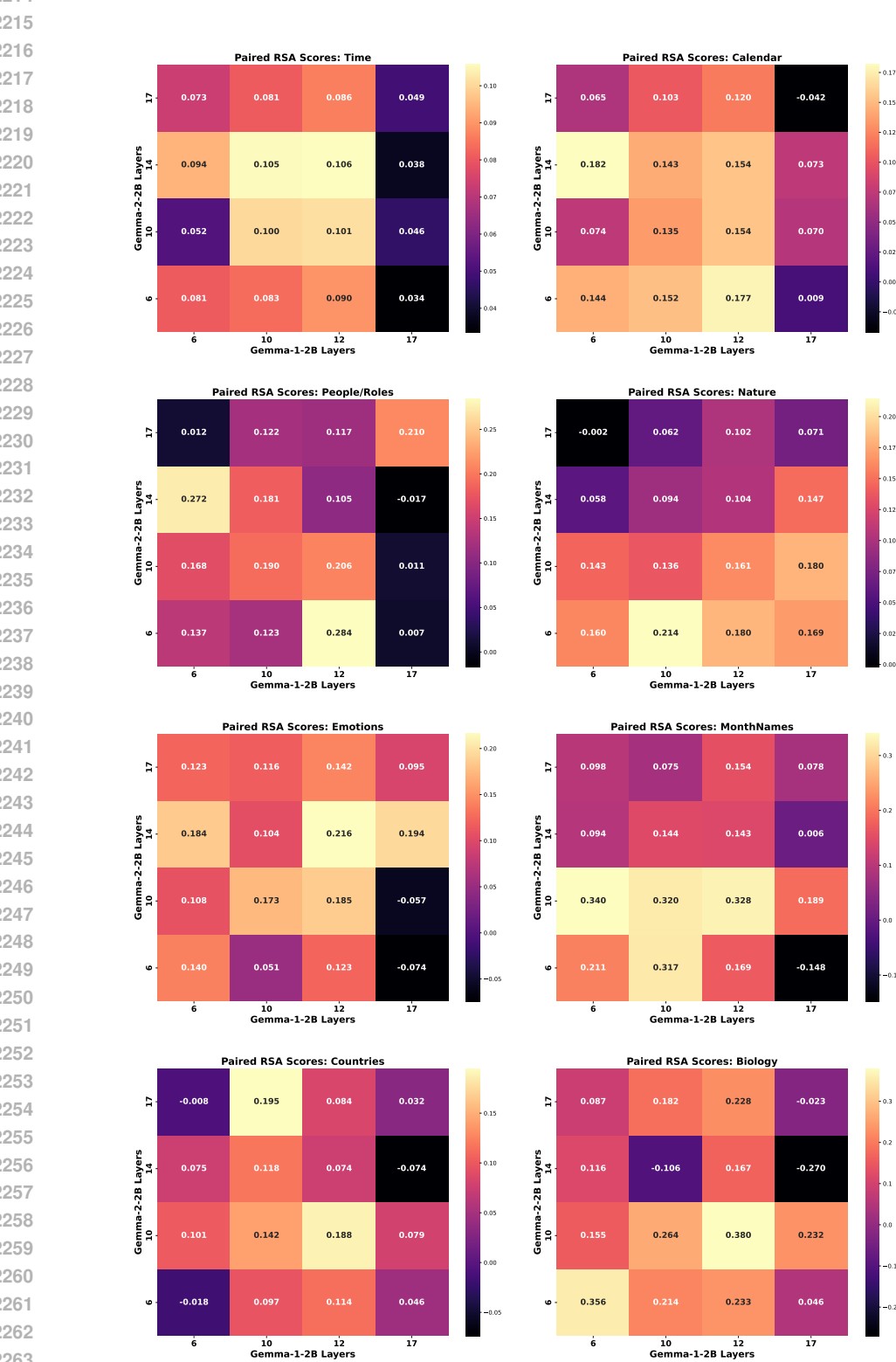

Figure 36: 1-1 Paired RSA scores of SAEs for layers in Gemma-1-2b vs layers in Gemma-2-2b for Concept Categories. Middle layers appear to be the most similar with one another.

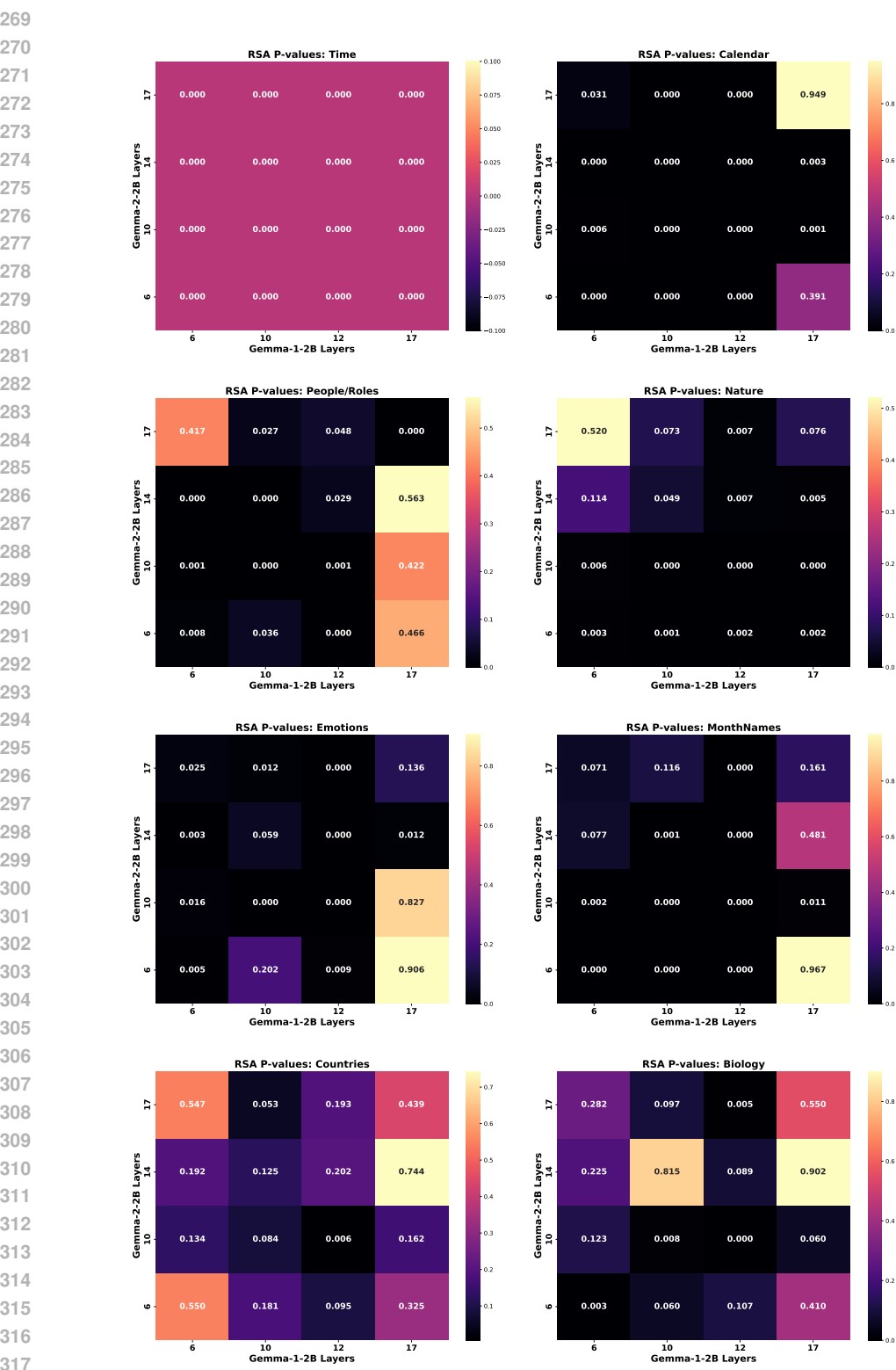

Figure 37: 1-1 RSA p-values of SAEs for layers in Gemma-1-2b vs layers in Gemma-2-2b for Concept Categories. A lower p-value indicates more statistically meaningful similarity.

