# OpenReview forum: "Measuring Sparse Autoencoder Feature Space Similarities Across Large Language Models"
_ICLR.cc/2026/Conference — ICLR 2026 Conference Withdrawn Submission_

### Official Review · Reviewer_Vgvs · 2025-10-30

**Soundness:** 3
**Presentation:** 2
**Contribution:** 2
**Rating:** 4
**Confidence:** 3

**Summary:**

This paper investigates the Universality Hypothesis in large language models (LLMs) through the lens of Sparse Autoencoders (SAEs).
Specifically, it proposes the idea of Analogous Feature Universality — even if SAEs trained on different models learn different feature representations, their feature spaces may be geometrically similar up to rotation-invariant transformations.
The authors pair features across SAEs using activation correlations, and then quantify space similarity via representational similarity measures such as SVCCA and RSA.
Empirically, they find that middle layers of diverse LLMs exhibit strong geometric alignment in their SAE feature spaces, suggesting partial universality of feature geometry across models.
The results imply that steering vectors or intervention directions might be transferrable across models through suitable transformations.

**Strengths:**

- Addresses universality, a key open concept in mechanistic interpretability.

- Demonstrates that SAE feature spaces align geometrically across models, suggesting latent structural regularities beyond specific feature identities.

- Introduces a rotation-invariant comparison framework (SVCCA, RSA on paired decoder weights), advancing methodological rigor.

- Empirical findings that middle layers exhibit highest universality are consistent with prior steering and alignment literature.

- The implication that steering directions could be transferable across models is conceptually exciting and potentially impactful for model control and safety.

**Weaknesses:**

- The claim of “Analogous Feature Universality” may be somewhat self-evident from prior universality work.
- It remains unclear how this extends beyond feature-level correlation findings (e.g., Crosscoder[1], Universal SAE[2]). The discussion of related work is relatively brief; clearer differentiation would help.
- The idea that SAE feature spaces can be aligned through rotational transformations, potentially allowing transfer of steering vectors or intervention directions across models, is very interesting.
However, is there any plan or direction for experiments to actually test such transferability?
If such experiments were conducted, the paper would be significantly stronger.

[1] Sparse Crosscoders for Cross-Layer Features and Model Diffing.
[2] Universal Sparse Autoencoders: Interpretable Cross-Model Concept Alignment.

**Questions:**

- Could the authors expand on why middle layers exhibit stronger universality?
- To what extent do your results depend on the dataset used for activation pairing (OpenWebText vs RedPajama)? Figure 16–17 suggests small differences; could these be formalized statistically?

---

### Official Review · Reviewer_LUHG · 2025-10-30

**Soundness:** 1
**Presentation:** 2
**Contribution:** 1
**Rating:** 2
**Confidence:** 4

**Summary:**

This paper aims to provide a methodology to measure similarity between SAE feature spaces across multiple LLMs, with the aim of providing evidence for the Universality Hypothesis, that different models converge towards similar concept representations in their latent spaces.

They propose a methodology which involves pairing feature directions from different SAEs, and computing a rotation-invariant similarity score between them, using either SVCCA or RSA.

The authors apply their methodology to SAE features learned on pairs of models from the same family of different sizes.

**Strengths:**

This paper aims to solve an important question, and makes a reasonable attempt to set up a solution.

**Weaknesses:**

Much of the wording in this paper is vague and inaccurate.

For example, in line 058, the authors suggest that "the spaces spanned by SAE features are similar" if and only if "one SAE space is similar to another SAE space under certain rotation-invariant transformations". Comparing the geometry of two point clouds up to orthogonal transformations is a standard and sensible thing to do. If this is what is intended, much clear description is required. In addition, this is inaccurate since the SAE features likely span the whole space.

The description of the pairing and pruning process is also hard to follow, and its seems very ad-hoc in nature and there is little to convince me that it is very robust.

While the two similarity measures are rotation invariant, I'm not convinced they are meaningful measures of similarity of SAE features. Firstly, the authors do not explain why they use SVCCA (i.e. CCA on the top eigenvectors) over CCA. Why should we perform CCA on the top eigenvectors and not the features as a whole? That said, CCA is not a good measure of similarity in this context. Two point clouds can have very different geometry and still have a high CCA score simply by having similar projections onto given direction.

In the statistical tests, multiple testing is not accounted for. Given that tests are performed on each pair of features this is significant.

The goal of the paper is to test fro universality of SAE feature spaces, however the only comparisons are between the same models of different sizes. Given that smaller versions of models are often derived directly from their larger counterparts, this is far from providing and evidence for the Universality Hypothesis. Is there a reason that experiments aren't performed across models?

Overall, I'm not convinced that the methodology meaningfully measures SAE feature space similarity.

**Questions:**

Please comment on the weaknesses mentioned.

What dimension $k$ is used in SVCCA in the experiments? Why is this value chosen?

---

### Official Review · Reviewer_p4Fn · 2025-10-31

**Soundness:** 3
**Presentation:** 3
**Contribution:** 3
**Rating:** 6
**Confidence:** 3

**Summary:**

The paper explores the universality of sparse autoencoders (SAEs) features across language models through a new lens, namely **Analogous Feature Universality**. Specifically, analogous feature universality hypothesizes that SAEs trained on different models learn feature spaces with similar underlying geometric structures, which remain comparable under rotation-invariant transformations. To this end, the paper proposes a two-step approach: first, they pair features across models using activation correlations; then, they compare the corresponding SAE decoder feature representations (the columns of the decoder/dictionary matrices) under rotation-invariant similarity measures to assess feature-space alignment. Empirical analyses reveal consistently high similarity scores across multiple language models, providing strong evidence for shared, universal structure in SAE feature spaces.

**Strengths:**

1. **Novel Approach:** The paper presents an original and well-motivated approach to studying the universality of SAE features across language models. The proposed method proceeds in two stages: first, features are paired across models using activation correlations; then, the corresponding SAE decoder feature representations (columns of the decoder or dictionary matrices) are compared under rotation-invariant similarity measures to assess feature-space alignment. This methodological design thoughtfully addresses the permutation problem by avoiding forced one-to-one mappings and by introducing a robust filtering process that excludes non-concept features and distinguishes between one-to-one and many-to-one pairings. However, the paper’s design and filtering choices implicitly acknowledge that **universality is selective, emerging only in specific, semantically coherent subspaces of the SAE feature space, not globally across every feature dimension**. Clarifying this interpretation would avoid misinterpretation of the findings.

2. **Comprehensive Empirical Evaluation:** The study evaluates eight pairs of models spanning multiple sizes and families, showing that the observed feature-space similarities consistently generalize across architectures and scales, reinforcing the robustness of the results.

3. **Writing Clarity:**
The paper is generally well-written and accessible, presenting complex methodological ideas with clarity. Some minor improvements could be made to streamline exposition and clarify methodological assumptions, as noted in the weaknesses section.

**Weaknesses:**

1. **Concern on Filtered Linear Similarity Metrics:** The paper’s similarity analysis relies on rotation-invariant linear measures (SVCCA and RSA) applied after substantial filtering of feature pairs. The authors note that only 10–30% (typically around 20%) of feature pairs are retained after removing non-concept and many-to-one mappings. While this filtering appropriately improves signal quality, it also means that the reported similarities reflect a selective and well-aligned subset of the feature space rather than its entirety. Combined with the reliance on linear metrics, this may overstate the extent of universality by emphasizing linearly aligned subspaces. Clarifying how universality trends change under less restrictive filtering or with nonlinear measures would strengthen the robustness of the conclusions.

2. **Lack of Empirical Validation:** The paper’s abstract suggests that Analogous Feature Universality could enable transferring interpretability techniques, such as steering vectors, across models. However, this implication is not empirically tested. Including even a preliminary experiment demonstrating such transferability would have strengthened the paper’s practical relevance and connection between geometric similarity and functional interpretability.

3. **Writing issues:** While the methodological description is conceptually clear, the paper would benefit from a more formal presentation using mathematical notation. For instance, lines 85–86 describe the approach as “a novel approach to study similar feature spaces by comparing weight representations, instead of activation representations, via paired features.” However, the term “weight” may be misinterpreted as referring to model parameters rather than SAE decoder feature representations. Such ambiguities could be avoided through the inclusion of precise notations and definitions.

**Questions:**

1. Is there a possibility that the filtered-out feature pairs contain any meaningful or conceptually relevant features? If so, how might their exclusion affect the strength or interpretation of your claim regarding feature-space universality?
2. How was the directionality of pairing chosen, and does this choice systematically affect the results?
3. How do the authors interpret low-similarity layers or models? Do these indicate representational divergence or methodological limits of the metrics?

---

### Note · Authors · 2026-01-06

I have read and agree with the venue's withdrawal policy on behalf of myself and my co-authors.